# SUBTASK-AWARE VISUAL REWARD LEARNING FROM SEGMENTED DEMONSTRATIONS

**Changyeon Kim**[1]  **Minho Heo**[1]  **Doohyun Lee**[1]
**Honglak Lee**[2,3]  **Jinwoo Shin**[1]  **Joseph J. Lim**[1*]  **Kimin Lee**[1*]
[1]KAIST  [2]University of Michigan  [3]LG AI Research

## ABSTRACT

Reinforcement Learning (RL) agents have demonstrated their potential across various robotic tasks. However, they still heavily rely on human-engineered reward functions, requiring extensive trial-and-error and access to target behavior information, often unavailable in real-world settings. This paper introduces REDS: *REward learning from Demonstration with Segmentations*, a novel reward learning framework that leverages action-free videos with minimal supervision. Specifically, REDS employs video demonstrations segmented into subtasks from diverse sources and treats these segments as ground-truth rewards. We train a dense reward function conditioned on video segments and their corresponding subtasks to ensure alignment with ground-truth reward signals by minimizing the Equivalent-Policy Invariant Comparison distance. Additionally, we employ contrastive learning objectives to align video representations with subtasks, ensuring precise subtask inference during online interactions. Our experiments show that REDS significantly outperforms baseline methods on complex robotic manipulation tasks in Meta-World and more challenging real-world tasks, such as furniture assembly in FurnitureBench, with minimal human intervention. Moreover, REDS facilitates generalization to unseen tasks and robot embodiments, highlighting its potential for scalable deployment in diverse environments.

## 1 INTRODUCTION

Reinforcement Learning (RL) has demonstrated significant potential for training autonomous agents in various real-world robotic tasks, provided that appropriate reward functions are available (Levine et al., 2016; Gu et al., 2017; Andrychowicz et al., 2020; Smith et al., 2023; Handa et al., 2023). However, reward engineering typically requires substantial trial-and-error (Booth et al., 2023; Knox et al., 2023) and extensive task knowledge, often necessitating specialized instrumentation (*e.g.*, motion trackers (Peng et al., 2020) or tactile sensors (Yuan et al., 2023)) or detailed information about target objects (James et al., 2020; Zhu et al., 2020; Yu et al., 2020; Mu et al., 2021; Gu et al., 2023; Sferrazza et al., 2024), which are difficult to obtain in real-world settings. Learning reward functions from action-free videos has emerged as a promising alternative, as it avoids the need for detailed action annotations or precise target behavior information, and video data can be easily collected from online sources (Soomro et al., 2012; Kay et al., 2017; Damen et al., 2018). Approaches in this domain include learning discriminators between video demonstrations and policy rollouts (Chen et al., 2021; Yang et al., 2024), training temporally aligned visual representations from large-scale video datasets (Sermanet et al., 2018; Zakka et al., 2021; Kumar et al., 2022; Ma et al., 2023b;a) to estimate reward based on distance to a goal image, and using video prediction models to generate reward signals (Escontrela et al., 2023; Huang et al., 2024).

Despite this progress, existing methods often struggle with long-horizon, complex robotic tasks that involve multiple subtasks. These approaches typically fail to provide context-aware reward signals, relying only on a few consecutive frames or the final goal image without considering subsequent subtasks. For example, in One Leg task (see Figure 2d) from FurnitureBench (Heo et al., 2023), prior methods often overemphasize the reward for picking up the leg while neglecting crucial steps such as inserting the leg into a hole and tightening it. Recent work (Mu et al., 2024) proposes a

---

*Equal advising.

**Project page:** https://changyeon.site/reds

discriminator-based approach that treats complex tasks as a sequence of subtasks. However, it assumes that the environment provides explicit subtask identification, which often demands significant human intervention in real-world scenarios. Moreover, discriminator-based methods are known to be prone to mode collapse (Wang et al., 2017; Zolna et al., 2021) (please refer to Figure 11 for empirical evidence over prior work). Consequently, designing an effective visual reward function for real-world, long-horizon tasks remains an open problem.

**Our approach**   To address the aforementioned limitations, we propose a novel reward learning framework, REDS: *REward learning from Demonstration with Segmentations*, which infers subtask information from video segments and generates corresponding reward signals for each subtask. The key idea is to employ minimal supervision to produce appropriate reward signals for intermediate subtask completion. Specifically, REDS utilizes expert demonstrations, where subtasks are annotated at each timestep by various sources (e.g., human annotators, code snippets, vision-language models; see the left figure of Figure 1). These annotations serve as ground-truth rewards. For training, we introduce a new objective function minimizing the Equivalent-Policy Invariant Comparison (EPIC) (Gleave et al., 2021) between the learned reward function and the ground-truth rewards, guaranteeing a theoretical upper bound on regret relative to the ground-truth reward function. Additionally, to correctly infer the ongoing subtask in online interactions, we adopt a contrastive learning objective to align video representations with task embeddings. In terms of architecture, our reward model is designed to capture temporal dependencies in video segments using transformers (Vaswani et al., 2017), leading to enhanced reward signal quality.

We find that REDS can generate appropriate reward signals to solve complex tasks by recognizing subtask structures, enabling the agent to efficiently explore and solve tasks through online interactions, using only expert demonstrations and subtask segmentations. Our experiments show that RL agents trained with REDS achieve substantially improved sample efficiency compared to baseline methods on various robotic manipulation tasks from Meta-World (Yu et al., 2020). Additionally, we show that REDS can effectively train agents to perform long-horizon, complex furniture assembly tasks from FurnitureBench (Heo et al., 2023) using real-world online RL. Moreover, REDS facilitates RL training in unseen environments involving new tasks and embodiments, which would otherwise require significant effort in prior reward-shaping methods.

**Contributions**   We highlight the key contributions of our paper below:

- We present a novel visual reward learning framework REDS: *REward learning from Demonstration with Segmentations*, which can produce suitable reward signals aware of subtasks in long-horizon complex robotic manipulation tasks.

- We show that REDS significantly outperforms baselines in training RL agents for robotic manipulation tasks in Meta-world, and even surpasses dense reward functions in some tasks.

- We demonstrate that REDS can train real-world RL agents to perform long-horizon complex furniture assembly tasks from FurnitureBench.

- We demonstrate that our approach shows strong generalization across various unseen tasks, embodiments, and visual variations.

## 2   RELATED WORK

**Reward learning from videos**   Learning from observations without expert actions has been a promising research area because it does not require extensive instrumentation and allows for the easy collection of vast amounts of video from online sources. Notably, several studies have proposed methods for learning rewards directly from videos and using the signal to train RL agents. Previous work has been focused on learning a reward function by aligning video representations in temporal order (Sermanet et al., 2018; Zakka et al., 2021; Kumar et al., 2022) while others train a reward function for expressing the progress of the agent towards the goal (Hartikainen et al., 2020; Lee et al., 2021; Yang et al., 2024). Most recent work (Escontrela et al., 2023) inspired by the success of video generative models (Yan et al., 2021; Ho et al., 2022) utilizes the likelihood of pre-trained video prediction models as a reward. To effectively utilize video for long-horizon tasks, we propose a new reward model conditioned both on video segments and corresponding subtasks trained with subtask segmentations.

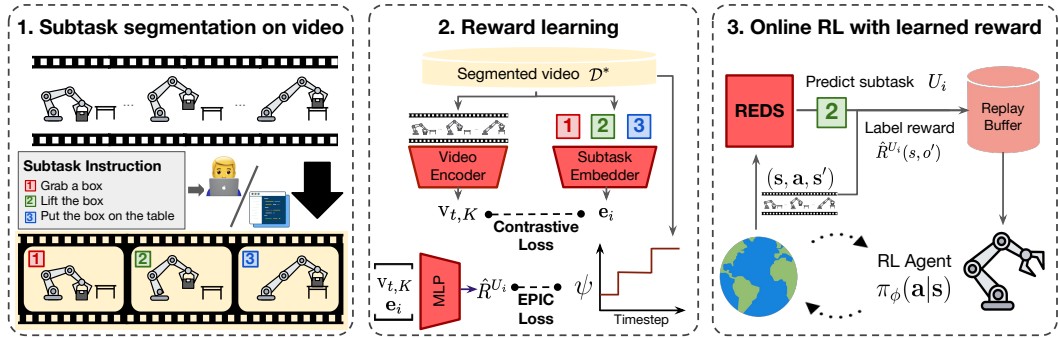

Figure 1: Illustration of REDS. Our main idea is to leverage expert demonstrations annotated with the ongoing subtask as the source of implicit reward signals (left). We train a reward model conditioned on video segments and corresponding subtasks with 1) contrastive loss to attract the video segments and corresponding subtask embeddings and 2) EPIC (Gleave et al., 2021) loss to generate reward equivalent to subtask segmentations (middle). In online RL, REDS infers the ongoing subtask using only video segments at each timestep and computes the reward with that (right).

**Inverse reinforcement learning** Designing an informative reward function remains a long-standing challenge for training RL agents. To achieve this, Inverse Reinforcement Learning (IRL) (Ng & Russell, 2000; Abbeel & Ng, 2004; Ziebart et al., 2008) aims to estimate the underlying reward function from expert demonstrations. Adversarial imitation learning (AIL) approaches (Ho & Ermon, 2016; Fu et al., 2018; Zolna et al., 2020; 2021; Mu et al., 2024) address this by training a discriminator network to discriminate transitions from expert data or policy rollouts and using the output from the discriminator as a reward for training agents with RL. The most similar work to ours is DrS (Mu et al., 2024), which also utilizes subtask information of the multi-stage task. While DrS assumes that the information on ongoing subtasks can be obtained from the environment during online interaction, our method has no such assumption, so it can be applied in more general cases when the segmenting of the subtask is hard in automatic ways (e.g., Heo et al. (2023)).

## 3 PRELIMINARIES

**Problem formulation** We formulate a visual control task as a Markovian Decision Process (MDP) (Sutton & Barto, 2018). As a single image observation is not sufficient for fully describing the underlying state of the task, we use the set of consecutive past observations to approximate the current state following common practice (Mnih et al., 2015; Yarats et al., 2021; 2022). Taking this into account, we define MDP as a tuple $\mathcal{M} = (\mathcal{S}, \mathcal{A}, p, R, \rho_0, \gamma)$. $\mathcal{S}$ is a state space consisting of a stack of $K$ consecutive images,

$\mathcal{A}$ is an action space, $R$ is the sparse reward function which outputs 1 when the agent makes success; otherwise, 0, $p(s'|s, a)$ is the transition function, $\rho_0$ is the initial state distribution, and $\gamma$ is the discount factor. The policy $\pi : \mathcal{S} \to \Delta(\mathcal{A})$ is trained to maximize the expected sum of discounted rewards $\mathbb{E}_{\rho_0, \pi, p} \left[ \sum_{t=0}^{\infty} \gamma^t R(s_t, a_t) \right]$. Our goal is to find a dense reward function $\hat{R}(s')$ only conditioned on visual observations, from which we can get the optimal policy $\pi^*$ for $\mathcal{M}$.

**EPIC** Equivalent-Policy Invariant Comparison (EPIC) (Gleave et al., 2021) is a pseudometric for quantifying differences between different reward functions, which is designed to ensure the invariance on the equivalent set of reward functions inducing the same set of optimal policies. To this end, EPIC first canonicalizes potential shaping of the reward function $R$ with some arbitrary distribution $\mathcal{D}_\mathcal{S} \in \Delta(\mathcal{S})$ over states $\mathcal{S}$, which is to be invariant to potential shaping as below[1]:

$$C_{\mathcal{D}_\mathcal{S}}(R)(s) = R(s) + \mathbb{E}_{S \sim \mathcal{D}_\mathcal{S}}[\gamma R(S) - R(S) - \gamma R(S)] = R(s) - \mathbb{E}_{S \sim \mathcal{D}_\mathcal{S}}[R(S)], \quad (1)$$

where $S$ denotes a set of batches independently sampled from the arbitrary distribution $\mathcal{D}_\mathcal{S}$, and $\gamma$ is the discount factor. EPIC is then defined by the Pearson distance between canonically shaped rewards in a scale-invariant manner:

$$D_{\mathcal{D}_\mathcal{C}, \mathcal{D}_\mathcal{S}}^{\text{EPIC}}(R_A, R_B) = \mathbb{E}_{s \sim \mathcal{D}_\mathcal{C}}\left[ D_\rho(C_{\mathcal{D}_\mathcal{S}}(R_A)(s), C_{\mathcal{D}_\mathcal{S}}(R_B)(s)) \right], \quad (2)$$

---

[1]We only consider the action-independent reward functions, and omit the prime notation on $s'$ for the simplicity of notation.

where $s$ is from the coverage distribution $\mathcal{D}_\mathcal{C}$, $D_\rho(X, Y) = \sqrt{\frac{1-\rho(X,Y)}{2}}$ is the Pearson distance between two random variables $X$ and $Y$, and $\rho(X, Y)$ is the Pearson correlation between $X$ and $Y$. Please refer to Gleave et al. (2021) for more details.

# 4 METHOD

This section presents REDS: *REward learning from Demonstration with Segmentations*, a visual reward learning framework designed for long-horizon tasks involving multiple subtasks. To generate proper reward signals for solving intermediate subtasks, we utilize segmentations identifying ongoing subtasks in demonstrations. In Section 4.1, we explain our intuition and formal definitions behind subtask segmentation. Section 4.2 outlines the reward model architecture, and Section 4.3 describes our training objective. Finally, in Section 4.3, we elaborate on the details of training and inference of REDS. For an overview, see Figure 1.

## 4.1 SUBTASK SEGMENTATION

The sparse reward function $R$ provides feedback only on the overall success or failure of a task, which is insufficient for guiding the agent through intermediate states. To address this, drawing inspiration from previous work on long-horizon robotic manipulation tasks (Di Palo & Johns, 2022; Mandlekar et al., 2023; Heo et al., 2023; Mu et al., 2024), we decompose a task into $m$ object-centric subtasks, denoted as $\mathcal{U} = \{U_1, ..., U_m\}$. Each subtask $U_i$ represents a distinct step in the task sequence and is based on the coordinate frame of a single target object. [2] and (ii) pulling the door to the goal position (which involves motion relative to the green sphere-shaped goal). This approach is intuitive because humans naturally perceive tasks as sequences of discrete object interactions, and this assumption can be generally applied to different manipulation skills (e.g., pick-and-place, inserting) with diverse objects. Additionally, we provide text instructions $\mathcal{X} = \{\boldsymbol{x}_i\}_{i=1}^m$ that describe how to solve each subtask, which helps guide the agent more effectively.

To obtain subtask segmentations, we map each observation $o_t$ at timestep $t$ in the trajectory $\tau = (o_0, ..., o_T)$ to its corresponding subtask using a segmentation function $\psi : \mathcal{O} \to \mathcal{U}$. Specifically, $\psi$ outputs the index of the ongoing subtask based on the observation at each timestep, with the output value increasing as the number of completed subtasks increases (refer to the graph of $\psi$ in the center of Figure 1). The function $\psi$ can be derived from various sources such as code snippets based on domain knowledge (James & Davison, 2022; James et al., 2022; Mees et al., 2022), guidance from human teachers (Heo et al., 2023), or vision-language models (Zhang et al., 2024; Kou et al., 2024). In our experiments, we use the predefined codes in Meta-world and human annotators in FurnitureBench to collect subtask segmentations. Note that these segmentations are only used during training; our framework is designed to automatically infer subtasks during online interactions without external annotations.

## 4.2 ARCHITECTURE

As mentioned in Section 1, previous reward learning methods generate rewards only by a single frame or consequent frames, not taking into account the order of subtasks. To resolve the issue, we propose a new reward predictor $\hat{R}^U = \hat{R}(s; U)$ conditioned on each subtask. To efficiently process visual observations, we first encode each image into low-dimensional representations using a pre-trained visual encoder $E_v$. To capture temporal dependencies, these representations are processed through a causal transformer (Vaswani et al., 2017). We add positional embeddings for each image in the sequence $s_t$ and pass them through the transformer network, producing the output representation $\boldsymbol{v}_{t,K} = \{\boldsymbol{v}_{t-K-1}, ..., \boldsymbol{v}_{t-1}, \boldsymbol{v}_t\}$ such that $t$-th output depends on input up to $t$. To embed subtask $U_i$, we encode $\boldsymbol{x}_i$ with pre-trained text encoder $E_t$ and project it to a shallow MLP to earn $\boldsymbol{e}_i$. This design allows REDS to generate rewards for unseen tasks when $\mathcal{U}$ and $\mathcal{X}$ are provided. (see Section 5.4 for supporting experiments). Finally, we concatenate a sequence of video representations $\boldsymbol{v}_{t,K}$ and subtask embedding $\boldsymbol{e}_i$ to $[\boldsymbol{v}_{t-K-1}, ...\boldsymbol{v}_t, \boldsymbol{e}_i]$ and project it to another shallow MLP $f$ to obtain $\hat{R}_\theta(s_t; U_i) = f(\boldsymbol{v}_{t,K}, \boldsymbol{e}_i)$.

---

[2]For instance, Door Open can be divided into (i) reaching the door handle (which involves motion relative to the door handle.)

### 4.3 REWARD MODELING

**Reward equivariance with subtask segmentation** Our key insight is that the subtask segmentation function $\psi$ can be thought of as the ground-truth reward function, providing implicit signals for solving intermediate tasks. To ensure our reward function induces the same set of optimal policies as $\psi$, we train to minimize EPIC (Gleave et al., 2021) distance between our reward model $\hat{R}_\theta^U$ parameterized by $\theta$ and $\psi$ for all subtasks:

$$\mathcal{L}_{\text{EPIC}}(\theta) = \frac{1}{k} \sum_{i=1}^{k} D_{\mathcal{D}_\mathcal{C}, \mathcal{D}_\mathcal{S}}^{\text{EPIC}}(\hat{R}_\theta^{U_i}, \psi). \tag{3}$$

**Progressive reward signal** However, minimizing EPIC with $\psi$ alone can lead to overfitting and the inability to provide progressive signals within each subtask. To mitigate this issue, we propose an additional regularization term to enforce progressive reward signals. Inspired by previous work (Lee et al., 2021; Hartikainen et al., 2020; Wu et al., 2021), we view the reward function as a progress indicator for each subtask, and we regularize the reward function output to be higher in later states of expert demonstration as follows:

$$\mathcal{L}_{\text{reg}}(\theta) = \max\left(0, \epsilon - (\hat{R}_\theta(s_{t+j}; \psi(o_{t+j})) - \hat{R}_\theta(s_t; \psi(o_t)))\right), \tag{4}$$

where $j$ is randomly chosen from a fixed set of values, and $\epsilon$ is a hyperparameter. Note that we apply this objective only for the expert demonstrations and not suboptimal demonstrations collected in iterative processes (please refer to Section 4.4).

**Aligning video representation with subtask embeddings** As the reward model lacks information about the ongoing subtasks in online interactions, it must infer the agent's current subtask. To achieve this, we train the video representation to be closely aligned with the corresponding subtask embedding by adopting a contrastive learning objective to make the model select the appropriate subtask embedding only by the video segment.

$$\mathcal{L}_{\text{cont}}(\theta) = -\log \frac{\text{sim}(\boldsymbol{v}_t, \boldsymbol{e}_{\psi(o_t)})}{\sum_{i \in \{1,...,k\}} \text{sim}(\boldsymbol{v}_t, \boldsymbol{e}_i)}, \tag{5}$$

where $\text{sim}$ represents a cosine similarity.

In summary, all components parameterized by $\theta$ are jointly optimized to minimize our total training objective:

$$\mathcal{L}(\theta) = \mathcal{L}_{\text{EPIC}}(\theta) + \mathcal{L}_{\text{reg}}(\theta) + \mathcal{L}_{\text{cont}}(\theta). \tag{6}$$

### 4.4 TRAINING AND INFERENCE

**Inference** To compute the reward at timestep $t$, REDS first infers the agent's current subtask from a sequence of recent visual observations. Specifically, REDS selects the subtask index $\bar{i}$ by choosing the subtask embedding $\boldsymbol{e}_{\bar{i}}$ that has the highest cosine similarity with the final output of the causal transformer, denoted as $\boldsymbol{v}_t$. The final reward is then computed using video embedding and text embedding of the inferred subtask as $\hat{R}_\theta(s_t; U_{\bar{i}}) = f(\boldsymbol{v}_t, \boldsymbol{e}_{\bar{i}})$. Please refer to Appendix A for more details.

**Training** We outline the training procedure for REDS. First, we collect subtask segmentations from expert demonstrations, creating a dataset $\mathcal{D}^0$, and use it to train the initial reward model, $M^0$. However, reward models trained solely on expert data are susceptible to reward misspecification (Pan et al., 2022). To address this, we iteratively collect suboptimal demonstrations and fine-tune the reward model using expert and suboptimal data. Unlike expert demonstrations, suboptimal demonstrations cover a broader range of states and more diverse observations, making manual segmentation labor-intensive and error-prone. To reduce the burden on human annotators, we develop an automatic subtask inference procedure, avoiding the need for manual segmentation.

Before the iterative process, we compute similarity scores for all states in the expert demonstrations using the initial reward model $M^0$. For each subtask $U_i$, we calculate a threshold $T_{U_i}$ based on the similarity scores between the expert states and the corresponding instructions, ensuring $T_{U_i}$ represents the minimum similarity required for successful subtask completion. In each iteration $i \in \{1, ..., n\}$, we proceed as follows:

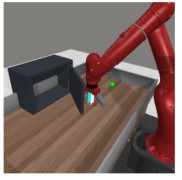 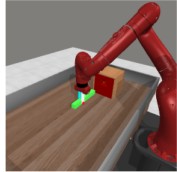 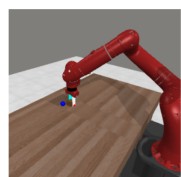 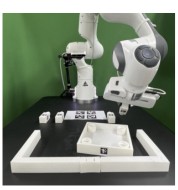 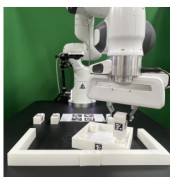

(a) Door Open    (b) Peg Insert Side    (c) Sweep Into        (d) One Leg

Figure 2: Examples of visual observations used in our experiments. We consider a variety of robotic manipulation tasks from Meta-world (Yu et al., 2020) and FurnitureBench (Heo et al., 2023).

- Step 1 (Suboptimal data collection): We train an RL agent using the reward model $M^i$ and collect suboptimal demonstrations $\mathcal{D}^i_{\text{replay}}$ from the agent's replay buffer.

- Step 2 (Subtask inference for suboptimal data): For each timestep in the suboptimal trajectory, we infer the subtask index $\hat{i}$ using the same procedure as in inference and compute $\text{sim}(\boldsymbol{v}_t, \boldsymbol{e}_{\hat{i}})$. If the similarity falls below the threshold $T_{U_i}$ at any timestep, we mark the subtask as failed and assign the remaining timesteps to that subtask.

- Step 3 (Fine-tuning): We fine-tune the reward model $M^{i-1}$ using the combined dataset $\mathcal{D}^i = \mathcal{D}^i \cup \mathcal{D}^i_{\text{replay}}$ to obtain $M^i$.

We use the final reward model $M^n$ for downstream RL training.

## 5 EXPERIMENTS

We design our experiment to evaluate the effectiveness of REDS on providing useful reward signals in training various RL algorithms (Hafner et al., 2023; Kostrikov et al., 2022). We conduct extensive experiments in robotic manipulation tasks from Meta-world (Yu et al., 2020) (see Section 5.1) in simulation and robotic furniture assembly tasks from FurnitureBench (Heo et al., 2023) (see Section 5.2) in the real-world. We also conduct in-depth analyses to validate the effectiveness of each component and how our reward function aligns with subtask segmentations (see Section 5.5).

**Implementation and training details** We used the open-source pre-trained CLIP (Radford et al., 2021a) with ViT-B/16 architecture to encode images and subtask instructions for all experiments. We adopt a GPT (Radford et al., 2018) architecture with 3 layers and 8 heads for the causal transformer. To canonicalize our reward functions, we use the same $\mathcal{D}$ for both coverage distribution $\mathcal{D}_{\mathcal{C}}$ and potential shaping distribution $\mathcal{D}_{\mathcal{S}}$, and we estimate the expectation over state distributions using a sample-based average over 8 additional samples from $\mathcal{D}$ per sample. All models are trained with AdamW (Loshchilov & Hutter, 2019) optimizer with a learning rate of $1 \times 10^{-4}$ and a mini-batch size of 32. To ensure to visual distractions, we apply color jittering and random shift (Yarats et al., 2021) to visual observations in training REDS. Please refer to Appendix A for more details.

**Baselines** We consider the following baselines: (1) human-engineered reward functions provided in the benchmark, (2) ORIL (Zolna et al., 2020), an adversarial imitation learning (AIL) method trained only with offline demonstrations, (3) Rank2Reward (R2R) (Yang et al., 2024), an AIL method which trains a discriminator weighted with temporal ranking of video frames to reflect task progress, (4) VIPER (Escontrela et al., 2023), a reward model utilizing likelihood from a pre-trained video prediction model as a reward signal, and (5) DrS (Mu et al., 2024), an AIL method that assumes subtask information from the environment and trains a separate discriminator for each subtask. We provide additional details on baselines in Appendix C.

### 5.1 META-WORLD EXPERIMENTS

**Setup** We first evaluate our method on 8 different visual robotic manipulation tasks from Meta-world (Yu et al., 2020). As a backbone algorithm, we use DreamerV3 (Hafner et al., 2023), a state-of-the-art visual model-based RL algorithm that learns from latent imaginary rollouts. For collecting subtask segmentations, we utilize a scripted teacher in simulation environments for scalability. Specifically, we use the predefined indicator for subtasks provided in the benchmark for all subtask segmentations (see Appendix D for the list of subtasks and corresponding text instructions for each task). We do not use these indicators when training/evaluating RL agents. For training

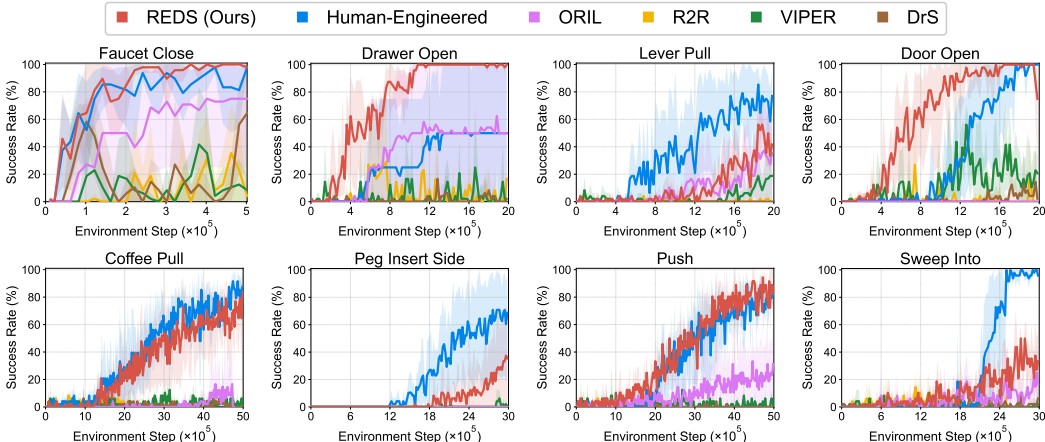

Figure 3: Learning curves of DreamerV3 (Hafner et al., 2023) agents trained with different reward functions for solving eight robotic manipulation tasks from Meta-world (Yu et al., 2020), measured by success rate (%). The solid line and shaded regions represent the mean and stratified bootstrap interval across 4 runs.

REDS, we first collect subtask segmentations from 50 expert demonstrations for initial training and train DreamerV3 agents for 100K environment steps with the initial reward model to collect suboptimal trajectories, which is used for fine-tuning. In evaluation, we measure the success rate averaged over 10 episodes in every 20K steps. Please refer to Appendix A for more details.

**Results** Figure 3 shows that REDS consistently improves the sample-efficiency of DreamerV3 agents by outperforming all baselines. While baselines exhibit non-zero success rates in simple tasks like Faucet Close, their performance significantly deteriorates in more complex tasks, such as Peg Insert Side. On the other hand, our method maintains non-zero success rates across all tasks and even surpasses human-engineered reward functions in some tasks (e.g., Drawer Open, Push, Coffee Pull) without requiring task-specific reward engineering. These results show that REDS effectively generates appropriate rewards for solving intermediate tasks by leveraging subtask-segmented demonstrations. A key advantage of REDS is that it relies solely on visual observations for generating rewards during online interaction, whereas DrS and human-engineered rewards require additional information from the environment, such as the position and reachability of target objects. This result underscores REDS's potential for application in environments where reward engineering is challenging or additional sensory information is unavailable.

## 5.2 FURNITUREBENCH EXPERIMENTS

**Setup** We further evaluate our method on real-world furniture assembly tasks from FurnitureBench (Heo et al., 2023), specifically focusing on One Leg assembly. This task involves a sequence of complex subtasks such as picking up, inserting, and screwing (see Figure 2d). For training REDS, we use 300 expert demonstrations with subtask segmentations provided by FurnitureBench, along with an additional 200 rollouts from IQL (Kostrikov et al., 2022) policy trained

Table 1: Online fine-tuning results of IQL agents in One Leg from FurnitureBench. We report the initial performance after offline RL (left) and the performance after 150 episodes of online RL (right).

| Method | # Expert Demos | Completed Subtasks (Offline → Online) |
|---|---|---|
| Sparse (Offline) (Heo et al., 2023) | 500 | 1.8 |
| VIPER | 300 | 1.10 → 1.25 |
| DrS | 300 | 1.05 → 1.10 |
| **REDS (Ours)** | **300** | **1.10 → 2.45** |

with expert demonstrations in a single training iteration. To prevent misleading reward signals stemming from visual occlusions, we utilize visual observations from the front camera and wrist cameras in training REDS. For downstream RL, we first train offline RL agents using 300 expert demonstrations labeled with each reward model, followed by online fine-tuning to assess improvements. For baselines, we compare against VIPER and DrS. We emphasize that our method enables fully autonomous training in online RL sessions, in contrast to DrS, which relies on a subtask indicator provided by humans. In our DrS experiments, subtasks were manually identified by a human. We measure the average number of completed subtasks over 20 rollouts for evaluation. We provide more details in Appendix A.

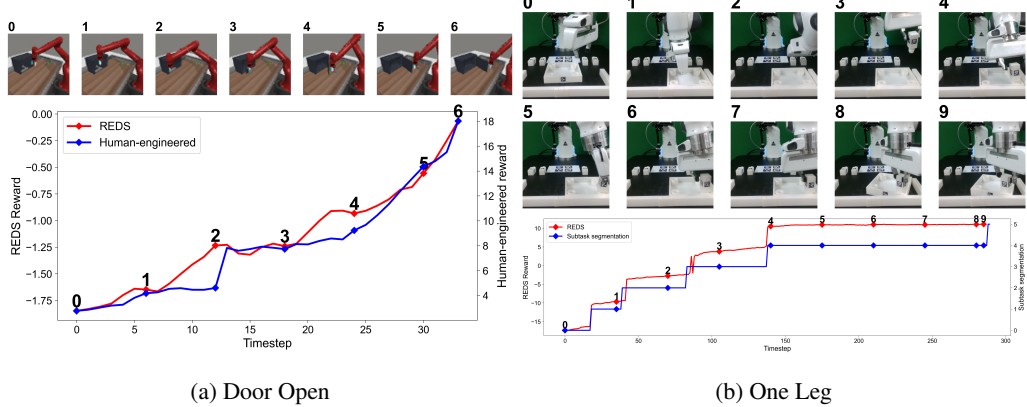

(a) Door Open            (b) One Leg

Figure 4: Qualitative results of REDS in Door Open in Meta-world (Yu et al., 2020) and One Leg from FurnitureBench (Heo et al., 2023). We observe that REDS produces suitable reward signals aligned with ground-truth reward functions by predicting ongoing subtasks effectively and providing progressive reward signals.

**Results** As shown in Table 1, REDS achieves significant performance improvements through online fine-tuning, whereas the improvements from baselines are marginal. These results indicate that our method produces informative signals for solving a sequence of subtasks, while baselines either fail to provide context-aware signals or dense rewards for better exploration (see Appendix F for qualitative examples). Moreover, we note that our method outperforms the IQL trained with 500 expert demonstrations, achieving a score of 2.45 compared to 1.8 reported by FurnitureBench, despite using only 300 expert demonstrations. Considering that REDS does not require additional human interventions beyond resetting the environment, these results highlight the potential to extend our approach to a wider range of real-world robotics tasks.

## 5.3 ALIGNMENT WITH GROUND-TRUTH REWARDS

**EPIC measurement** To quantitatively validate the alignment of our method with ground-truth reward functions, we measure the EPIC distance with a set of unseen demonstrations during training. Specifically, we use rollouts from the reference policy trained with expert demonstrations for state distribution. In Table 2, we observe that REDS exhibits significantly lower EPIC distance than baselines across all tasks. Particularly, the difference be-

Table 2: EPIC (Gleave et al., 2021) distance (lower is better) between learned reward functions and hand-engineered reward functions (Meta-world) / subtask segmentations (FurnitureBench) in unseen data.

| Task | VIPER | R2R | ORIL | REDS (Ours) |
|---|---|---|---|---|
| Meta-world Door Open | 0.5934 | 0.5649 | 0.7071 | **0.4913** |
| Meta-world Push | 0.6144 | 0.6838 | 0.7073 | **0.5381** |
| Meta-world Peg Insert Side | 0.5974 | 0.5806 | 0.6989 | **0.4674** |
| Meta-world Sweep Into | 0.6248 | 0.6413 | 0.7001 | **0.4673** |
| FurnitureBench One Leg | 0.7035 | 0.6001 | 0.7014 | **0.0713** |

tween REDS and baselines is more pronounced in complex tasks like One Leg. This result consistently supports the empirical findings from previous sections.

**Qualitative analysis** We provide the graph of computed rewards from REDS in Figure 4. We observe that REDS can induce suitable reward signals aligned with ground-truth reward functions. For example, REDS provides subtask-aware signals in transition states (e.g., between 2 and 3, and between 4 and 5) and generates progressive reward signals throughout each subtask. Please refer to Appendix F for the extensive comparison between REDS and baselines.

## 5.4 GENERALIZATION CAPABILITIES

**Transfer to unseen tasks** As mentioned in Section 4.2, our model can be applied as a reward function in unseen tasks. To validate this, we conduct additional experiments by training REDS with segmentation data from 3 tasks (Door Open, Drawer Open/Close) and using the reward model to train RL agents in two unseen tasks. In Door Close, we aim to validate that REDS can provide informative signals for a new task involving a previously seen object and behaviors. In Window Close, we aim to determine whether REDS can provide suitable reward signals for familiar behaviors (closing) with an unseen object (window). In evaluation, we change the text instruction following

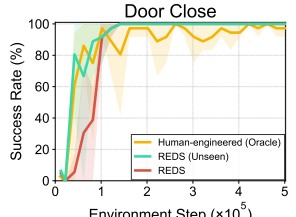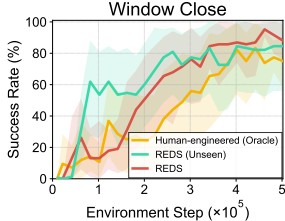

Figure 5: We train REDS with 3 different tasks from Meta-world (Yu et al., 2020) and use this model to train RL agents in 2 unseen tasks (left). We present learning curves on Door Close (center) and Window Close (right), as measured by success rate (%). The solid line and shaded regions represent the mean and stratified bootstrap interval across 4 runs.

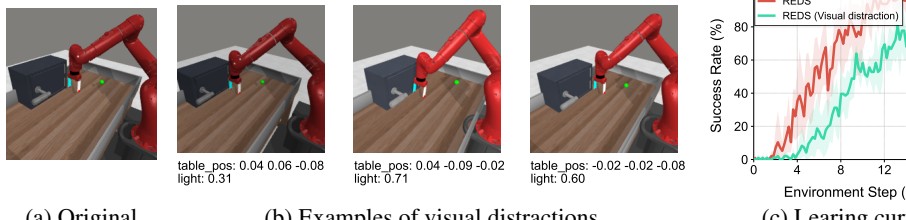

(a) Original  (b) Examples of visual distractions  (c) Learing curve

Figure 6: We provide visual observations from (a) the original environment and (b) unseen environments with visual distractions used in our experiments in Section 5.4 .

the target object (as shown in Table D), and we do not fine-tune the reward model. Figure 5 shows that REDS provides effective reward signals on unseen tasks and achieves comparable or even better RL performance than REDS trained on the target task. This result demonstrates that REDS can be applied to RL training in unseen tasks that share properties with training tasks.

**Robustness to visual distractions**  To prove the robust performance of REDS against visual distractions, we train RL agents with our reward model in new Meta-world environments incorporating visual distractions, such as varying light and table positions following Xie et al. (2024) (see Figure 6b). Note that the reward model was trained using demonstrations only from the original environment. As Figure 6c shows, REDS can generate robust reward signals despite visual distractions and train RL agents to solve the task effectively.

**Transfer to unseen embodiments**  Since our framework leverages only action-free video data, we hypothesize that transferring to other robot embodiments with similar DoFs is feasible. To support this claim, we train REDS with demonstrations of the Franka Panda Arm and then compute the reward of an unseen demonstration of the Sawyer Arm in Take Umbrella Out of Stand from RL-Bench (James et al., 2020). Figure 8 shows that REDS generates informative reward signals even with the unseen embodiment. For instance, REDS can capture the behavior of taking the umbrella out of the stand, as indicated by the increased reward signals between 6 and 7. Additionally, Figure 7 shows that REDS trained only with the Panda Arm can be used to train downstream RL agents in the environment with the Sawyer Arm.

Figure 7: Learning curve for DreamerV3 agents in environments of the Sawyer Arm.

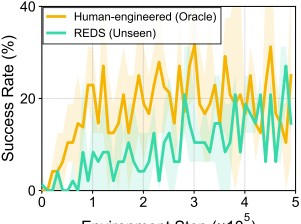

## 5.5 ABLATION STUDIES

**Effect of training objectives**  We investigate the effect of each training objective in Figure 9a. Specifically, we compare REDS with 1) a baseline trained with regression to subtask segmentation instead of EPIC loss $\mathcal{L}_{\text{EPIC}}$, 2) a baseline that utilizes only video representations without subtask embeddings, and 3) a baseline trained without the regularization loss $\mathcal{L}_{\text{reg}}$. We observe that RL performance significantly degrades without each component, implying that our losses synergistically improve reward quality.

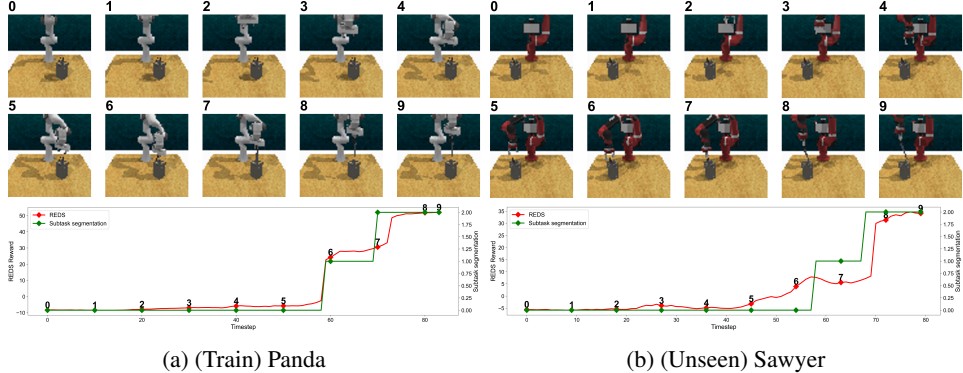

| (a) (Train) Panda | (b) (Unseen) Sawyer |

Figure 8: Qualitative results of REDS with different robot embodiments. REDS was trained using demonstrations from the Panda Arm and evaluated on an unseen demonstration from the Sawyer Arm in Take Umbrella Out of Stand from RLBench (James et al., 2020). We visualize several frames above the graph and mark them with a diamond symbol.

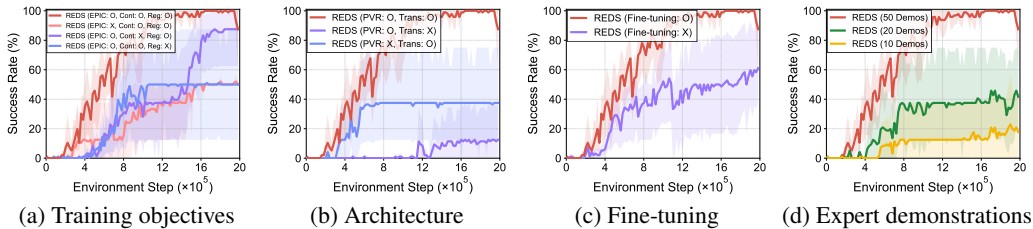

| (a) Training objectives | (b) Architecture | (c) Fine-tuning | (d) Expert demonstrations |

Figure 9: Learning curves for two Meta-world (Yu et al., 2020) robotic manipulation tasks, measured by success rate (%), to examine the effects of (a) training objectives, (b) architecture, (c) fine-tuning, and (d) the number of expert demonstrations. The solid line and shaded regions show the mean and stratified bootstrap interval across 8 runs.

**Effect of architecture**    To verify the design choice proposed in Section 4.2, we compare REDS with 1) a baseline using a CNN for encoding images instead of pre-trained visual representations (PVR) and 2) a baseline simply concatenating pre-trained visual representations without a causal transformer. Figure 9b shows that both baselines show worse performance compared to ours. Notably, detaching a causal transformer significantly degrades RL performance, implying that temporal information is essential for providing suitable reward signals in robotic manipulation.

**Effect of fine-tuning**    In Figure 9c, we compare REDS trained only with the expert demonstrations in the initial phase to REDS fine-tuned with additional suboptimal demonstrations as described in Section 4.4. REDS shows improved RL performance when trained with additional suboptimal demonstrations, indicating that the coverage of state distribution impacts the reward quality. Further investigation on how to efficiently collect suboptimal demonstrations to enhance the performance of learned reward function is a promising future direction.

**Effect of the number of expert demonstrations**    We investigate the effect of the number of expert demonstrations by measuring the RL performance of DreamerV3 agents with REDS trained with different numbers of expert demonstrations in 2 tasks (Door Open, Drawer Open) from Meta-world. Figure 9d shows that the agents' RL performance positively correlates with the number of expert demonstrations trained for reward learning.

## 6    CONCLUSION

We proposed REDS, a visual reward learning framework considering subtasks by utilizing subtask segmentation. Our main contribution is based on proposing a new reward model leveraging minimal domain knowledge as a ground-truth reward function. Our approach is generally applicable and does not require any additional instrumentations in online interactions. We believe REDS will significantly alleviate the burden of reward engineering and facilitate the application of RL to a broader range of real-world robotic tasks.

## ACKNOWLEDGEMENTS

This research is supported by Institute of Information & Communications Technology Planning & Evaluation (IITP) grant funded by the Korea government (MSIT) (RS-2022-II220953, Self-directed AI Agents with Problem-solving Capability; RS-2019-II190075 Artificial Intelligence Graduate School Program (KAIST); No. RS-2024-00509279, Global AI Frontier Lab).

## LIMITATION AND FUTURE DIRECTIONS

One limitation of our work is that we assume the knowledge of the object-centric subtasks in a task. For automating subtask definition and segmentation in new tasks and new domains other than robotic manipulations, investigating the planning and reasoning capabilities of pre-trained Multimodal Large Language Model (MLLM) (Park et al., 2023; Honerkamp et al., 2024; Zawalski et al., 2024; Shah et al., 2024; Liu et al., 2024) would be an intriguing research direction.

Additionally, the performance of REDS relies on pre-trained representations trained with natural image/text data for encoding videos and subtask instructions. Although REDS proves its effectiveness in various robotic manipulation tasks, we observe that REDS struggles to distinguish subtle changes (e.g., screwing the leg in One Leg) even with pre-trained representations trained on ego-centric motion videos (Ma et al., 2023a). We believe that the quality of rewards can be further improved by utilizing 1) pre-trained representations with large-scale data with diverse robotic tasks (Padalkar et al., 2023; Khazatsky et al., 2024) and 2) representations trained with objectives considering affordances (Bahl et al., 2023) or object-centric methods (Devin et al., 2018).

Furthermore, there is room for improvement to enhance generalization and robustness. Although our experiments are designed to evaluate generalization in unseen environments, they may face challenges in out-of-distribution environments, such as significant changes in the background or camera angles. Future work could address these challenges through data augmentation or domain adaptation techniques. while our contrastive learning objective currently focuses on minimizing distances between relevant video and text embeddings, the reward model may generate inappropriate reward signals for semantically different subtasks that share similar video content and text instructions. Incorporating a loss term to maximize distances between irrelevant embeddings could further improve robustness in tasks with similar subtasks, and we will explore this enhancement in future work.

Finally, the number of expert demonstrations and the number of iterations for fine-tuning REDS are determined by empirical trials. Investigating how to collect failure demonstrations to mitigate reward misspecification efficiently is an interesting future direction.

## ETHIC STATEMENT

Video demonstrations and subtask segmentations used in the experiments were sourced from publicly available benchmarks (Meta-world, RLBench, FurnitureBench), ensuring no personal or sensitive information is involved. Potential risks could arise when training and deploying RL agents directly in real-world scenarios, particularly in human-robot interactions. Ensuring the safety and reliability of these agents before deployment is essential to prevent harm.

## REPRODUCIBILITY STATEMENT

For the reproducibility of REDS, we have provided a detailed explanation of implementation details and experimental setups in Section 4.4, Section 5, and Appendix A. In addition, to further facilitate the reproduction, we release the open-sourced implementation through the project website.

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

## A EXPERIMENT DETAILS

**Training and inference details** To ensure robustness against visual changes, we apply data augmentations, including random shifting (Yarats et al., 2021; 2022) and color jittering. For optimization, we train REDS with AdamW (Loshchilov & Hutter, 2019) optimizer with a learning rate of $1 \times 10^{-4}$, weight decay of $2 \times 10^{-2}$, and a cosine decay schedule for adjusting the training learning rate. We apply a warm-up scheduling for the initial 500 gradient steps starting from a learning rate of 0. Note that the parameters for CLIP visual/text encoders have not been updated. For training downstream RL agents, we normalize the reward by dividing it by the maximum value observed in the expert demonstrations. We report the hyperparameters used in our experiments in Table 3. For both coverage distribution $\mathcal{D}_\mathcal{C}$ and potential shaping distribution $\mathcal{D}_\mathcal{S}$, we use the same dataset with subtask segmentations $\mathcal{D}^i$, unlike prior work dealing with arbitrarily random distributions because of the absence of subtask segmentations. To canonicalize our reward functions, we estimate the expectation over state distributions using a sample-based average over 8 additional samples from $\mathcal{D}$ per sample. To prevent false positive cases in predicting subtasks in online interactions, we add margins to similarity scores inversely proportional to the subtasks in online interactions. Specifically, we infer the subtask $\hat{i}$ as follows:

$$\hat{i} = \text{argmax}_{i \in \{1,...,k\}}(\text{sim}(\boldsymbol{v}_t, \boldsymbol{e}_i) + \eta * (k - i)), \tag{7}$$

where $\eta$ is a hyperparameter for the margin between subtasks. For each subtask $U_i$, we compute similarity scores between the visual observations within the subtask from expert demonstrations and their corresponding instructions. The threshold $T_{U_i}$ is set to the 75th percentile of these scores to account for demonstration variability while capturing the most relevant matches. Please refer to Figure 12c for supporting experiments.

Table 3: Hyperparameters of REDS used in our experiments.

| Hyperparameter | Value |
| --- | --- |
| Batch size | 32 (Meta-world, RLBench), 8 (FurnitureBench) |
| Training steps | 5000 |
| Learning rate | 0.0001 |
| Optimizer | AdamW (Loshchilov & Hutter, 2019) |
| Optimizer momentum | $\beta_1 = 0.9$, $\beta_2 = 0.999$ |
| Weight decay | 0.02 |
| Learning rate decay | Linear warmup and cosine decay |
| Warmup steps | 500 |
| Context length | 4 |
| Causal transformer size | 3 layers, 8 heads, 512 units |
| EPIC canonical samples | 8 |
| $\epsilon$ for progressive reward signal | 0.05 |
| $\eta$ for inferring subtasks | 0.01 (Meta-world, RLBench), 0.05 (FurnitureBench) |
| number of training iterations $n$ | 2 (Meta-world, RLBench), 1 (FurnitureBench) |

**Meta-world experiments** We use visual observations of $64 \times 64 \times 3$. To consistently use a single camera viewpoint over all tasks, we use the modified version of the `corner2` viewpoint as suggested by Seo et al. (2022). Expert demonstrations for each task are collected using scripted policies publicly released in the benchmark. We use an action repeat of 2 to accelerate training and set the maximal episode length as 250 for all Meta-world tasks. For downstream RL, we use the implementation of DreamerV3 from VIPER[3]. We report the hyperparameters of DreamerV3 agents used in our experiments in Table 4. Unless otherwise specified, we use the same set of hyperparameters as VIPER. For all ablation experiments (Figure 6, 9, 12), we report results in Door Open and Drawer Open.

**RLBench experiments** For training both reward models and downstream RL agents, we utilize $64 \times 64 \times 3$ RGB observations from the front camera and wrist camera. For downstream RL, we don't use any expert demonstrations, and we use the same set of hyperparameters as VIPER.

---

[3]https://github.com/Alescontrela/viper_rl

Table 4: Hyperparameters of DreamerV3 (Escontrela et al., 2023) used in Meta-world experiments.

| Hyperparameter | Value |
|---|---|
| **General** | |
| Replay Capacity (FIFO) | $5 \times 10^5$ |
| Start learning (prefill) | 5000 |
| MLP size | $2 \times 512$ |
| **World Model** | |
| RSSM size | 512 |
| Base CNN channels | 32 |
| Codes per latent | 32 |

Table 5: Hyperparameters of IQL (Kostrikov et al., 2022) used in FurnitureBench experiments.

| Hyperparameter | Value |
|---|---|
| Learning rate | $3 \times 10^{-4}$ |
| Batch size | 256 |
| Policy # hidden units | (512, 256, 256) |
| Critic/value # hidden units | (512, 256, 256) |
| Image encoder | R3M (Nair et al., 2022) |
| Discount factor ($\gamma$) | 0.996 |
| Expectile ($\tau$) | 0.8 |
| Inverse Temperature ($\beta$) | 10.0 |

**FurnitureBench experiments** We use the implementation of IQL from FurnitureBench [4] for our experiments. We utilize $224 \times 224 \times 3$ RGB observations from the front camera and wrist cameras, along with proprioceptive states, to represent the current state. We encode each image with pre-trained R3M (Nair et al., 2022) for visual observations. Following Kostrikov et al. (2022), we first run offline RL for 1M gradient steps, then continue training while collecting environment interaction data, adding it to the replay buffer, and repeating this process for 150 episodes. Before online fine-tuning, we pre-fill the replay buffer with 10 rollouts from the pre-trained IQL policy. We adopt techniques from RLPD (Ball et al., 2023) for efficient offline-to-online RL training. Specifically, we sample 50% of the data from the replay buffer and the remaining 50% from the offline data buffer containing 300 expert demonstrations. We also apply LayerNorm (Ba et al., 2016) in the critic/value network of the IQL agent to prevent catastrophic overestimation. We list the hyperparameters used in our experiments in Table 5. For training REDS, we collect subtask segmentations for suboptimal demonstrations using the automatic subtask inference procedure described in Section 4.4, and we manually modified some subtask segmentations with false negatives to guarantee stable performance.

**Computation** We use 24 Intel Xeon CPU @ 2.2GHz CPU cores and 4 NVIDIA RTX 3090 GPUs for training our reward model, which takes about 1.5 hours in Meta-world and 3 hours in FurnitureBench due to high-resolution visual observations from multiple views. For training DreamerV3 agents in Meta-world, we use 24 Intel Xeon CPU @ 2.2GHz CPU cores and a single NVIDIA RTX 3090 GPU, which takes approximately 4 hours over 500K environment steps. For training IQL agents in FurnitureBench, we use 24 Intel Xeon CPU @ 2.2GHz CPU cores and a single NVIDIA RTX 3090 GPU, taking approximately 2 hours for 1M gradient steps in offline RL and 4.5 hours over 150 episodes of environment interactions in online RL.

## B REDS ARCHITECTURE DETAILS

We encode visual observations with a pre-trained CLIP (Radford et al., 2021b) ViT-B/16 visual encoder, utilizing all representations from the sequence of patches. We adopt 1D learnable parameters with the same size for positional embedding, and we add these parameters to 2D fixed sin-cos embeddings and add them to features. To encode temporal dependencies in visual observations, we use a GPT (Radford et al., 2018) architecture with 3 layers and 8 heads. In FurnitureBench, we use a sequence of images from both the front camera and wrist camera as input. Given $s_t^{\text{front}}/s_t^{\text{wrist}}$ from the front/wrist camera, we concatenate visual observations to $[o_{t-K-1}^{\text{front}}, o_{t-K-1}^{\text{wrist}}, ..., o_t^{\text{front}}, o_t^{\text{wrist}}]$, add positional embeddings, 2D fixed sin-cos embeddings, and additional 1D learnable parameters for each viewpoint for effectively utilizing images from multiple cameras. We then pass the features to the transformer layer, the same as the model with a single image. The subtask embedder and final reward predictor are implemented as 2-layer MLPs.

---

[4] https://github.com/clvrai/furniture-bench/tree/main/implicit_q_learning

## C  BASELINE DETAILS

**ORIL (Zolna et al., 2020)**   For implementing ORIL with visual observations, we use the CNN architecture from Yarats et al. (2021) to encode image observations. For training data, we use the same set of demonstrations as for training REDS. Since our training data are divided into success and failure demonstrations, we do not use positive-unlabeled learning (Xu & Denil, 2021) in our experiments. For robustness against visual changes, we apply the same augmentation techniques used for training REDS.

**Rank2Reward (R2R) (Yang et al., 2024)**   To ensure compatibility with backbone RL algorithms (Hafner et al., 2023; Kostrikov et al., 2022) implemented in JAX, we reimplement the reward model with JAX following the official implementation of Rank2Reward [5] and use the same hyperparameters. We first pre-train the ranking network using the same expert demonstrations as REDS, and we then train a discriminator for the expert demonstration and policy rollouts, weighted by the output from the pre-trained ranking network. For training efficiency, we use the CNN architecture from Yarats et al. (2021) for encoding visual observations instead of R3M (Nair et al., 2022), finding no significant difference when we use the pre-trained visual representations like R3M, but with much slower training in online RL. We observe that our R2R implementation with DreamerV3 in JAX outperforms the original version implemented with DrQ-V2 (Yarats et al., 2022) agents.

**DrS (Mu et al., 2024)**   Similar to R2R, we reimplement DrS with JAX following the official implementation of DrS [6], and use the same set of hyperparameters for reward learning. As the original DrS implementation is based on a state-based environment, we switch the backbone RL algorithm from SAC to DrQ-V2 (Yarats et al., 2022) and apply the augmentation technique in the reward learning phase for processing visual observations efficiently. To report the RL performance, we use the learned dense reward model to train new RL agents. In FurnitureBench experiment, we train the reward model with the same expert/failure demonstrations as in Section 5.2, without online interaction, to avoid unsafe behaviors and a significant increase in training time from online interactions.

**VIPER (Escontrela et al., 2023)**   We use the official implementation of VIPER [7] for our experiments. Given the similarities among robotic manipulation tasks, we use the same set of hyperparameters as in RLBench (James et al., 2020) experiments to train VQ-GAN and VideoGPT. We train 100K steps, choosing the checkpoint with the minimum validation loss. In FurnitureBench experiment, we use images from the front camera, resized to $64 \times 64 \times 3$, and set the exploration objective $\beta$ as 0.

## D  TASK DESCRIPTIONS

In this section, we list the subtasks and corresponding text instructions for each task in Table 6. For Meta-world tasks, we provide the code snippet used to determine the success of each subtask (Please refer to the Meta-world (Yu et al., 2020) for more details). For the FurnitureBench One Leg task, we outline the criteria used by human experts to assess the success of each subtask based on the metric defined in FurnitureBench (Heo et al., 2023).

## E  EXTENDED RELATED WORK

**Quantifying differences between reward functions**   Previous work has explored methods for measuring the difference between reward functions without relying on policy optimization procedures (Gleave et al., 2021; Wulfe et al., 2022; Skalse et al., 2024). In particular, Gleave et al. (2021) introduced the EPIC distance, a pseudometric invariant to equivalent classes of reward functions. Subsequent work (Rocamonde et al., 2024; Adeniji et al., 2022; Liang et al., 2022) has employed EPIC to assess the quality of reward functions. In this paper, we take a different approach by using EPIC distance as an optimization objective. While Adeniji et al. (2022) also utilizes EPIC distance

---

[5] https://github.com/dxyang/rank2reward
[6] https://github.com/tongzhoumu/DrS
[7] https://github.com/Alescontrela/viper_rl

Table 6: A list of subtasks and language description for each subtask used for REDS in our experiments.

| Task | Subtask | Success condition | Language description |
|---|---|---|---|
| Meta-world Faucet Close | 1 | object_grasped ≤ 0.9 | a robot arm reaching the faucet handle. |
| | 2 | target_to_obj ≤ 0.07 | a robot arm rotating the faucet handle to the right. |
| Meta-world Drawer Open | 1 | gripper_error ≤ 0.03 | a robot arm grabbing the drawer handle. |
| | 2 | handle_error ≤ 0.03 | a robot arm opening a drawer to the green target point. |
| | 3 | handle_error ≤ 0.03 | a robot arm holding the drawer handle near the green target point after opening. |
| Meta-world Lever Pull | 1 | ready_to_lift > 0.9 | a robot arm touching the lever. |
| | 2 | lever_error ≤ np.pi/24 | a robot arm pulling up the lever to the red target point. |
| Meta-world Door Open | 1 | reward_ready ≥ 1.0 | a robot arm grabbing the door handle. |
| | 2 | abs(obs[4] − self._target_pos[0]) ≤ 0.08 | a robot arm opening a door to the green target point. |
| | 3 | abs(obs[4] − self._target_pos[0]) ≤ 0.08 | a robot arm holding the door handle near the green target point after opening. |
| Meta-world Coffee Pull | 1 | tcp_to_obj < 0.04 ∧ tcp_open > 0 | a robot arm grabbing the coffee cup. |
| | 2 | obj_to_target ≤ 0.07 | a robot arm moving the coffee cup to the green target point. |
| | 3 | obj_to_target ≤ 0.07 | a robot arm holding the cup near the green target point. |
| Meta-world Peg Insert Side | 1 | tcp_to_obj < 0.03 ∧ tcp_open > 0 | a robot arm grabbing the green peg. |
| | 2 | obj[2] − 0.1 > self.obj_init_pos[2] | a robot arm lifting the green peg from the floor. |
| | 3 | obj_to_target ≤ 0.07 | a robot arm inserting the green peg to the hole of the red box. |
| | 4 | obj_to_target ≤ 0.07 | a robot arm holding the green peg after inserting. |
| Meta-world Push | 1 | tcp_to_obj ≤ 0.03 | a robot arm grabbing the red cube. |
| | 2 | target_to_obj ≤ 0.05 | a robot arm pushing the grabbed red cube to the green target point. |
| | 3 | target_to_obj ≤ 0.05 | a robot arm holding the grabbed red cube near the green target point. |
| Meta-world Sweep Into | 1 | self.touching_main_object > 0 ∧ tcp_opened > 0 | a robot arm grabbing the red cube. |
| | 2 | target_to_obj ≤ 0.05 | a robot arm sweeping the grabbed red cube to the blue target point. |
| | 3 | target_to_obj ≤ 0.05 | a robot arm holding the grabbed red cube near the blue target point. |
| Meta-world Door Close | 1 | in_place == 1.0 | a robot arm grabbing the door handle. |
| | 2 | obj_to_target ≤ 0.08 | a robot arm closing a door to the green target point. |
| | 3 | obj_to_target ≤ 0.08 | a robot arm holding the door handle near the green target point after closing. |
| Meta-world Window Close | 1 | tcp_to_obj ≤ 0.05 | a robot arm grabbing the window handle. |
| | 2 | target_to_obj ≤ 0.05 | a robot arm closing a window from left to right. |
| | 3 | target_to_obj ≤ 0.05 | a robot arm holding the window handle after closing. |
| FurnitureBench One Leg | 1 | robot gripper tips make contact with one surface of the tabletop. | a robot arm picking up the white tabletop. |
| | 2 | nearest corner of the tabletop is placed close to the right edge of the obstacle. | a robot arm pushing the white tabletop to the front right corner. |
| | 3 | robot gripper securely grasps a leg of the table and lifts it. | a robot arm picking up the white leg. |
| | 4 | leg is inserted into one of the screw holes of the tabletop, and the robot releases the gripper. | a robot arm inserting the white leg into screw hole. |
| | 5 | leg is fully assembled to the tabletop. | a robot arm screwing the white leg until tightly lifted. |
| | 6 | leg is fully assembled to the tabletop. | a robot arm holding the white leg in place. |
| RLBench Take Umbrella Out of Umbrella Stand | 1 | GraspedCondition(self.robot.gripper, self.umbrella).condition_met()[0] | a robot arm grasping the umbrella. |
| | 2 | DetectedCondition(self.umbrella, self.success_sensor, negated = True).condition_met()[0] | a robot arm taking the grasped umbrella ouf of the umbrella stand. |
| | 3 | DetectedCondition(self.umbrella, self.success_sensor, negated = True).condition_met()[0] | a robot arm holding the umbrella on the umbrella stand. |

Table 7: A list of language description used for CLIP and LIV.

| Task | Language description |
|---|---|
| Meta-world Door Open | a robot arm grabbing the drawer handle and opening the drawer. |
| Meta-world Drawer Open | a robot arm grabbing the door handle and opening the door to the green target point. |

for optimizing intrinsic reward functions in skill discovery, our method applies EPIC distance to train dense reward functions for long-horizon tasks, serving as a direct reward signal for RL training.

**Segmenting demonstrations for long-horizon manipulation tasks** Several approaches have been proposed to decompose long-horizon demonstrations into multiple subgoals to prevent error accumulation and provide intermediate signals for agent training. These include extracting key points from proprioceptive states (James & Davison, 2022; James et al., 2022; Shridhar et al., 2022; Shi et al., 2023), employing greedy heuristics on off-the-shelf visual representations pre-trained with robotic data (Zhang et al., 2024), and learning additional modules on top of pre-trained visual-language models to align with keyframes (Kou et al., 2024). Our work builds on these efforts by leveraging subtask segmentations but focuses on developing a reward learning framework that explicitly incorporates subtask decomposition to generate suitable reward signals for intermediate tasks. Additionally, we further demonstrate that our model generalizes effectively to unseen tasks and robot embodiments.

## F    EXTENDED QUALITATIVE ANALYSIS

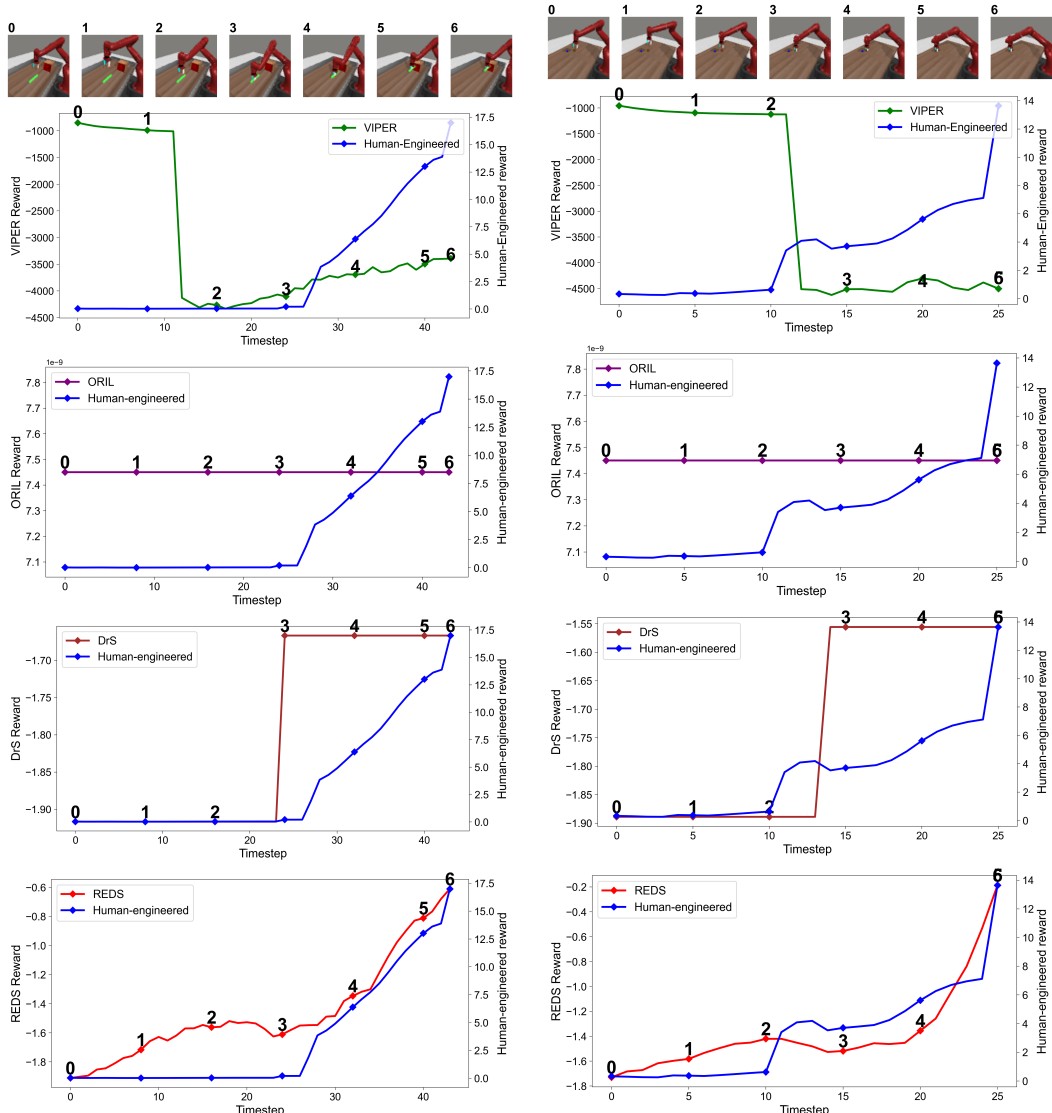

Figure 10: Qualitative results of VIPER (Escontrela et al., 2023), ORIL (Zolna et al., 2020), DrS (Mu et al., 2024), and REDS (Ours) in Peg Insert Side (left), and Sweep Into (right) from Meta-world Yu et al. (2020). We visualize several frames above the graph and mark them with a diamond symbol.

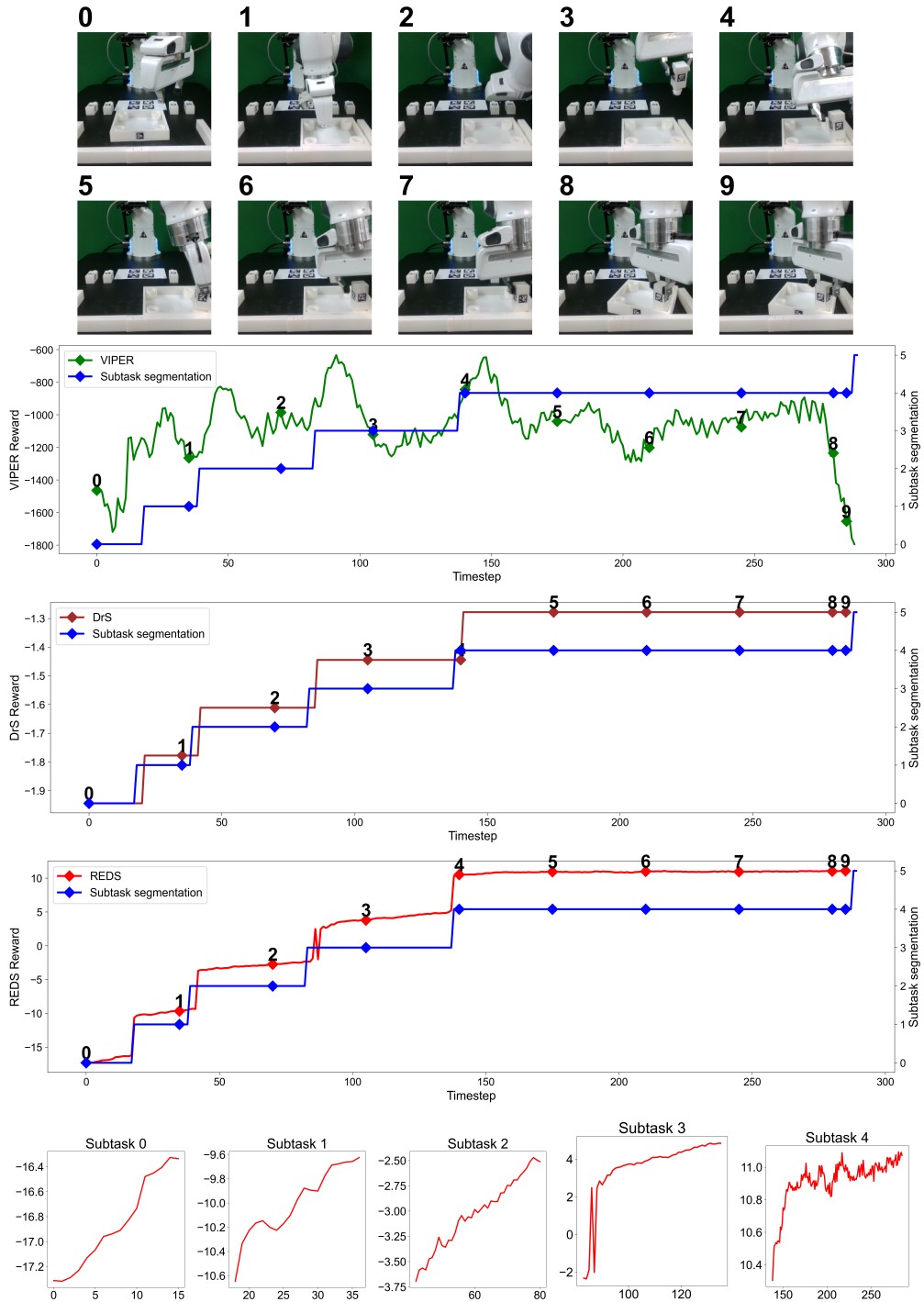

Figure 11: Qualitative results of VIPER (Escontrela et al., 2023), DrS (Mu et al., 2024), and REDS (Ours) in One Leg from FurnitureBench (Heo et al., 2023). We visualize several frames above the graph and mark them with a diamond symbol. VIPER, which does not utilize subtask information, assigns lower rewards to later subtasks, making agents stagnate in earlier phases. While DrS uses ground-truth subtask information from the environment, it produces sparse reward signals within each subtask. In contrast, REDS provides subtask-aware signals in subtask transitions and generates progressive reward signals (see the bottom figure zoomed in for each subtask).

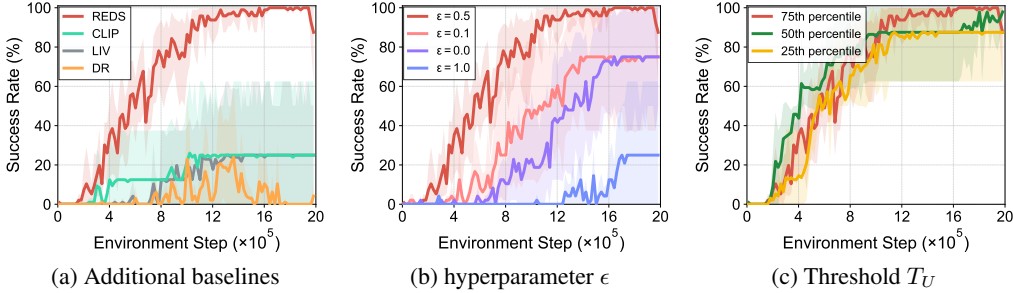

Figure 12: Learning curves for two Meta-world (Yu et al., 2020) robotic manipulation tasks, measured by success rate (%). The solid line and shaded regions show the mean and stratified bootstrap interval across 4 runs.

## G    ADDITIONAL EXPERIMENTS

**Comparison with additional baselines**    We first compare REDS with additional baselines, CLIP (Radford et al., 2021a) and LIV (Ma et al., 2023a), utilizing the distance between visual observation and text instructions for generating rewards. It's important to note that these models cannot infer subtasks in online interaction unlike REDS; therefore, we use other text instructions describing how to solve the whole task. (refer to Table 7 for details). Figure 12a shows that REDS significantly outperforms baselines, indicating that providing detailed signals aware of subtasks is crucial for better RL performance. Additionally, we compare REDS with Diffusion Reward (DR) (Huang et al., 2024), which utilizes conditional entropy from a video diffusion model as a reward signal. Our findings indicate that REDS also significantly outperforms DR. This is attributed to the fact that DR does not explicitly incorporate subtask information, which is essential for generating context-aware rewards in long-horizon tasks. These results further emphasize the advantage of REDS in handling tasks requiring precise subtask guidance.

**Effect of scaling progressive reward signals**    In Figure 12b, we examine the effect of $\epsilon$ scaling the regularization for progressive reward signals in Equation 4. We observe that $\epsilon = 0.5$ shows the best performance, while smaller values relatively weaken helpful progressive signals, and larger values degrade the reward function by reducing the accuracy in inferring subtasks.

**Effect of threshold $T_U$ for subtask inference**    We present experimental results using various threshold $T_U$ in Figure 12c. We observe that a lower percentile threshold exhibits lower RL performance. These results indicate that a lower percentile threshold allows more observations to be classified as successful; however, this can lead to misleading subtask identification, resulting in decreased RL performance.

**Subtask identification ability of REDS**    To assess the subtask identification capability of REDS, we measure its precision before and after fine-tuning with additional suboptimal demonstrations. This evaluation involves using 50 unseen expert demonstrations and 50 suboptimal demonstrations sampled from the replay buffer of DreamerV3 agents that were trained with a human-engineered reward. Table 8 shows that the precision is comparable for expert demonstrations for both agents; however, there

Table 8: Precision in identifying subtasks of REDS on 50 unseen expert demonstrations and 50 unseen suboptimal demonstrations.

| Fine-tuning | Expert | Suboptimal | Total |
|---|---|---|---|
| ✗ | 94.49% | 70.90% | 82.70% |
| ✓ | 92.56% | 91.49% | 92.03% |

is a significant increase in precision for suboptimal demonstrations after fine-tuning. This improvement in precision results in enhanced RL performance, as illustrated in Figure 9c.

**Additional EPIC measurements** To further validate the efficacy of our method, we measure the EPIC distance between learned reward functions and subtask segmentations in Meta-world environments. Note that we report the result with the same set of unseen demonstrations used in Section 5.3. In Table 9, we observe that REDS exhibits significantly lower EPIC distance than baselines across all tasks, consistently supporting the claims from the experiments in the main text.

Table 9: EPIC (Gleave et al., 2021) distance (lower is better) between learned reward functions and subtask segmentations in unseen data.

| Task | VIPER | R2R | ORIL | REDS (Ours) |
|---|---|---|---|---|
| Meta-world Door Open | 0.6017 | 0.5731 | 0.7017 | **0.4870** |
| Meta-world Push | 0.6293 | 0.7014 | 0.7094 | **0.5129** |
| Meta-world Peg Insert Side | 0.6384 | 0.6021 | 0.7001 | **0.4381** |
| Meta-world Sweep Into | 0.6179 | 0.6584 | 0.7011 | **0.4293** |

