# OpenReview forum: "Subtask-Aware Visual Reward Learning from Segmented Demonstrations"
_ICLR.cc/2025/Conference — ICLR 2025 Poster_

### Official Review · Reviewer_7C5k · 2024-10-28

**Soundness:** 3
**Presentation:** 3
**Contribution:** 3
**Rating:** 6
**Confidence:** 2

**Summary:**

This paper introduces REDS, a novel framework aimed at improving reward learning for reinforcement learning (RL) in complex robotic tasks. The framework tackles the problem of designing reward functions, which typically require extensive human effort and domain-specific knowledge. Instead of relying on predefined reward functions, REDS uses segmented video demonstrations where subtasks are labeled, allowing the model to learn from these segmentations with minimal supervision.

The key innovation lies in aligning video segments with subtasks using contrastive learning and employing the EPIC (Equivalent-Policy Invariant Comparison) distance to ensure that the learned reward function is consistent with ground-truth subtasks. The model is tested on both simulated tasks (Meta-World) and real-world tasks (FurnitureBench) and demonstrates superior performance in long-horizon tasks with multiple subtasks.

**Strengths:**

1. A novel framework for reward learning.
2. Experiments are very extensive, including both simulated and real-world tasks (FurnitureBench).

**Weaknesses:**

1. The method relies on pre-trained visual and text encoders (e.g., CLIP), which may not be optimal for robotic tasks. The paper acknowledges that subtle visual changes are not well-handled, and reward quality could be further improved with better pretraining.
2. While REDS reduces the need for hand-crafted reward functions, it still requires segmented demonstrations. The quality and quantity of these demonstrations could impact the performance.

**Questions:**

See weakness.

---

> ### Author Response · Authors · 2024-11-21
>
> Dear Reviewer 7C5k,
>
> We sincerely appreciate your valuable comments, which were extremely helpful in improving our draft. Below, we address each comment in detail.
>
> ---
>
> **[W1] Reliance on pre-trained visual and text encoders**
>
> We appreciate the reviewer’s insightful comment. We hypothesize that pre-trained models leveraging large-scale robotic datasets [1, 2] could improve reward quality by capturing subtle visual changes more effectively. However, to the best of our knowledge, no such open-sourced model is currently available. We consider this a promising avenue for future work and plan to investigate these representations as they become available.
>
> **[W2] Effect of segmented demonstrations**
>
> Thank you for your comment. While REDS requires demonstrations with subtask segmentations, we designed it to mitigate dependence on high-quality annotations. The iterative training process leverages suboptimal demonstrations with automatic segmentation, reducing the need for manual effort (Section 4.4). Moreover, as shown in Figure 9d, REDS performs robustly even with fewer demonstrations, indicating its adaptability to varying data quality.
>
> ---
>
> **Reference**\
> [1] Open X-Embodiment: Robotic Learning Datasets and RT-X Models, ICRA 2024.\
> [2] DROID: A Large-Scale In-the-Wild Robot Manipulation Dataset, RSS 2024.\
> [3] Robotic Control via Embodied Chain-of-Thought Reasoning, CoRL 2024.

---

> ### Author Response · Authors · 2024-11-24
> **Gentle Reminder**
>
> Dear Reviewer 7C5k,
>
> Thank you again for your time and efforts in reviewing our paper.
>
> As the discussion period draws close, we kindly remind you that two days remain for further comments or questions. We would appreciate the opportunity to address any additional concerns you may have before the discussion phase ends.
>
> Thank you very much!
>
> Many thanks,\
> Authors

---

> > ### Comment · Reviewer_7C5k · 2024-11-25
> > **Thank the authors for the reply**
> >
> > Thanks for the detailed responses and additional results. Most of my concerns are well addressed. Besides, same as other reviewers, I wish to see your further experiment results on more complex real-world robot tasks. I maintain my score as above acceptance threshold.

---

> > > ### Author Response · Authors · 2024-11-25
> > >
> > > Dear Reviewer 7C5k,
> > >
> > > Thank you for your feedback. We are pleased to hear that we have addressed most of your concerns. As per your suggestion, we will incorporate experimental results on more complex real-world robot tasks in the final manuscript. Please feel free to reach out with any additional questions or suggestions.
> > >
> > > Thank you once again!
> > >
> > > Many thanks,\
> > > The Authors

---

### Official Review · Reviewer_VAFb · 2024-10-31

**Soundness:** 3
**Presentation:** 4
**Contribution:** 3
**Rating:** 8
**Confidence:** 5

**Summary:**

This paper presents a novel visual reward learning framework that aligns video demonstration segments with learned rewards, effectively indicating progress in subtask completion. The performance improvements across eight simulation tasks and a real-world assembly task demonstrate the method's efficacy. The visualizations of the rewards, the framework's generalization capabilities, and the impacts of various design choices significantly enhance the comprehension of the learned reward.

**Strengths:**

- The paper is well-organized, providing clear descriptions of the proposed approach, including effective reward visualizations and experimental setups.
- Experiments are thorough and convincing. Results are favorable compared with SOTA approaches like VIPER and DrS. The comprehensive ablation study clearly delineates the impact of each component.
- The experiments on generalization ability and real robots are impressive. It is good to see these properties are held in the proposed method.

**Weaknesses:**

While I think this paper deserves acceptance based on its strong results and presentation, I have several concerns that should be addressed in the next revision:

- **Context-Aware Reward Signals**: While intuitively beneficial, learning context-aware reward signals may require extensive annotation, potentially weakening the motivation for this work.
    - Existing studies, such as VIP and XIRL, have successfully learned context-aware reward signals without context labels but are not compared in this paper.
    - The method bears resemblance to that presented in [1], which employs purely unsupervised learning without requiring subtasks; this similarity should be discussed and compared.
    - Although the authors suggest that large models could be used to alleviate annotation burdens, this claim is not substantiated in the current version. I recommend either removing this mention or including an experiment that compares large-model-based annotations with human-annotated ones.
- **Clarification of Motivation**: The motivation behind the proposed method needs more clarity. In Line 049, the authors reference prior approaches in the "One Leg Task," but this is not empirically supported in Figure 2 or subsequent sections.
- **Extension to Other Domains:** The applicability of the proposed method to other domains is unclear. Specific rules for subtask segmentation and the threshold $T_U$ should be explicitly described. The authors are encouraged to discuss how to adapt existing methods to other domains, what components need more attention, and potential failure scenarios.
- **More Experiments**: The experiments could be more convincing. While the authors claim that the proposed reward can better handle long-horizon tasks, the selected tasks are not too long-horizon, in my perspective. It is suggested to extend One Leg task to Two/Three Leg Tasks, and see the changes in learned reward and the corresponding performance.
- **Generalization Failures**: It is impressive to see experiments on generalization ability. However, it would be beneficial to present cases where the proposed method struggles. For instance, the authors could include discussions on scenarios such as changes in camera view or background.
- I also include several smaller questions below that should be addressed in the next revision as well.

**Questions:**

- What rules are used for the division of each task? Could different rules yield varying downstream performance?
- In Table 1, why do DrS, VIPER, and REDS perform worse than Sparse Reward after the offline phase (approximately 1.1 vs. 1.8)?
- What is the effect of $T_U$? Could the authors conduct experiments to explore this?
- In Table 2, while REDS achieves the highest EPIC score, what happens if hand-engineered rewards are replaced with sparse ones (e.g., step-style rewards like in the One Leg Task)? Would the conclusion remain valid?
- In Figure 4, REDS appears continuous in (a) but step-like in (b), despite both rewards being learned to minimize the distance between $\phi$. What accounts for this difference in appearance of the learned rewards?
- Why is Diffusion Reward not included in the experiments? It may outperform VIPER.
- Does the format of the language instruction influence the results?
- What explains the sudden drops for VIPER and DrS and flat curve of ORIL in Figure 10?

[1] Sermanet, P., Xu, K., & Levine, S. (2016). Unsupervised perceptual rewards for imitation learning. RSS, 2017.

---

> ### Author Response · Authors · 2024-11-21
>
> Dear Reviewer VAFb,
>
> We sincerely appreciate your valuable comments, which were extremely helpful in improving our draft. Below, we address each comment in detail.
>
> ---
>
> **[W1-1,1-2] Comparison with existing studies**
>
> VIP [1] and XIRL [2] share similarities with REDS in learning visual representations from demonstration videos and generating progressive reward signals. However, they require access to goal images to compute rewards, which is often impractical in real-world settings. In contrast, REDS autonomously generates reward signals without relying on additional environmental information, enabling fully autonomous online RL training.
>
> Sermanet et al. [3] propose an unsupervised approach to derive reward signals by discovering intermediate steps from visual features. This method has a limitation in that it depends entirely on expert demonstrations and pre-trained visual features, making it susceptible to reward misattribution in visually ambiguous scenarios (e.g., confusing leg insertion with alignment in the "One Leg" task). REDS addresses these limitations by leveraging subtask-segmented demonstrations to train additional models on top of pre-trained representations, offering precise, context-aware guidance in long-horizon and complex tasks. Furthermore, REDS effectively incorporates suboptimal demonstrations, mitigating reward misspecification and enhancing robustness.
>
> **[W1-3] Using large-scale models to alleviate annotation burdens**
>
> Thank you for pointing this out. Following your suggestions, we revised the draft to remove this mention and added an in-depth discussion on limitations and future works (please refer to page 11 of the revised draft).
>
> **[W2] Clarification of motivation**
>
> The proposed method's motivation stems from the limitations of prior approaches like VIPER, as shown in Figure 11. VIPER assigns lower rewards to later subtasks (beyond point 4), making agents stagnate in earlier phases. In contrast, our method generates context-aware reward signals that maintain consistency across subtasks, enabling effective learning for complex, long-horizon tasks. We have revised the manuscript to elaborate on this motivation and its empirical support.
>
> **[W3-1] Clarification on rules for subtask segmentation**
>
> To ensure consistent subtask definitions, we assume tasks comprise a sequence of object-centric subtasks, where each subtask involves manipulating a single object, following prior work [4,5]. For instance, Door Open (Figure 2a of the draft) can be divided into (i) reaching the door handle (which involves motion relative to a handle.) and (ii) pulling the door to the goal position (which involves motion relative to a green sphere). For new tasks, subtasks can be similarly defined by identifying changes in the target object, and this approach aligns with human intuition and is repeatable. We have not tried different rules and suspect that excessive task horizons without proper subtask decomposition can degrade RL performance. In the revised manuscript, we have expanded the discussion on subtask definition rules.
>
> **[W3-2] Clarification on rules for threshold $T_{U}$**
>
> For each subtask $U_i$, we compute similarity scores between the visual observations within the subtask from expert demonstrations and their corresponding instructions. The threshold ${T_{U_i}$ is set to the 75th percentile of these scores to account for demonstration variability while capturing the most relevant matches. We include this clarification in the revised manuscript.
>
> **[W3-3] Extension to other domains**
>
> Thank you for your valuable feedback. We agree that discussing the adaptability of REDS to other domains is important. REDS can be extended with minimal modifications, as its framework is inherently domain-agnostic. The primary considerations involve defining subtasks and decomposing expert demonstrations. To ensure robustness, the task horizon should be constrained to avoid excessively large state distributions, which could increase susceptibility to reward hacking. For example, in web control tasks, the problem can be decomposed into subtasks aligned with changes in target components (e.g., screens, icons, or scrollbars).
> Moreover, leveraging the reasoning capabilities of (Multimodal) Large Language Models (MLLMs) offers a scalable approach to subtask definition and demonstration decomposition. This has been demonstrated in domains such as navigation [6, 7], agent control [8, 9], and mobile manipulation [10, 11]. We will further clarify these points in the final manuscript, including a discussion of potential failure scenarios and domain-specific considerations, to strengthen the generalizability and applicability of REDS.

---

> > ### Author Response · Authors · 2024-11-21
> >
> > **[W4] Additional experiments with more complex tasks**
> >
> > We appreciate your insightful suggestion. While the current experiments are designed to evaluate the method’s ability to handle long-horizon tasks effectively, we agree that extending to more complex tasks, such as assembling multiple legs, would further strengthen the claims. To this end, we are currently collecting additional expert demonstrations for these extended tasks, as open-source demonstrations are insufficient for achieving task success. This data will enable us to train the reward model and evaluate its performance in more complex scenarios. We will update the manuscript with these results in future iterations.
> >
> > **[W5] Failure cases for generalization**
> >
> > While REDS demonstrates its ability to generalize effectively to unseen environments, we acknowledge that significant out-of-distribution shifts, such as significant changes in background and camera angles, may pose challenges. To address these, future work could explore advanced data augmentation techniques, such as synthetic variation in backgrounds and viewpoints, or domain adaptation strategies to enhance robustness. We will incorporate a detailed discussion of these limitations and potential extensions in the revised manuscript to provide a clear roadmap for addressing these challenges.
> >
> > **[Q2] Clarification on performances in Table 1**
> >
> > The observed performance difference can be attributed to the experimental setup. Specifically, the Sparse Reward results referenced from the original FurnitureBench paper [12] were obtained by training the IQL agent with 500 expert demonstrations. In contrast, our experiments utilize only 300 expert demonstrations for the initial offline RL phase.
> >
> > **[Q3] Effect of threshold $T_{U}$**
> >
> > Thank you for highlighting this important point. Following your suggestion, we conducted additional experiments to evaluate the effect of $T_U$, the threshold used for detecting failure points in suboptimal demonstrations. Figure 12c of the revised manuscript shows that the 75% percentile exhibits the best performance, and lower percentile thresholds result in reduced RL performance. We expect this is because lower thresholds classify a larger number of observations as successful, leading to incorrect subtask identification and hindering learning.
> >
> > **[Q4] EPIC score with sparse rewards**
> >
> > Thank you for your suggestion. Table 8 of the revised manuscript evaluates the EPIC distance between the learned reward function and the subtask identification function using the same set of unseen demonstrations used in Section 5.3. The results show that REDS achieves significantly lower EPIC distances than all baselines, even when sparse step-style rewards (as in the One Leg Task) are used instead of hand-engineered rewards. This demonstrates the robustness of REDS in aligning reward functions with task progress, consistently supporting the conclusions drawn from the experiments in the main text.
> >
> > **[Q5] Clarification on Figure 4**
> >
> > The difference in the appearance of the learned rewards arises due to the task horizon and the granularity of the reward visualization. In longer-horizon tasks, the learned reward may appear step-like in aggregated views. However, when visualized at a finer granularity, the reward trends exhibit progressive reward signals. To clarify this, we have included separate reward graphs for each subtask in Figure 11 of the revised manuscript, illustrating the progressive reward signals of the learned rewards within individual subtasks.
> >
> > **[Q6]  New baseline: Diffusion Reward**
> >
> > Diffusion Reward (DR) [13] is a concurrent method that utilizes conditional entropy from a video diffusion model as a reward signal for training RL agents. Following your suggestion, we conducted an additional comparison with DR. As shown in Figure 12a of the revised manuscript, REDS significantly outperforms DR. This is because DR does not explicitly incorporate subtask information, which is critical for generating context-aware rewards in long-horizon tasks. These results further highlight the advantage of REDS in handling tasks requiring precise subtask guidance.

---

> ### Author Response · Authors · 2024-11-21
>
> **[Q7] Influence of the form of language instructions**
>
> We observe that the format of language instructions has minimal impact on the performance of downstream RL agents, as the model primarily relies on the semantic content rather than specific syntactic structure. To ensure compatibility and consistency, we adopt the format used in CLIP, which is widely validated for language-vision tasks.
>
> **[Q8] Explanation of Figure 10**
>
> Figure 10 illustrates the limitations of VIPER, DrS, and ORIL in providing effective reward signals. ORIL suffers from mode collapse and fails to distinguish visual changes in the robot's state, resulting in improper reward signals. The sudden drop in DrS performance (right figure) is caused by misleading subtask annotations from the pre-defined script; we have corrected this issue and updated the graph in the revised draft. VIPER generates distinct reward scales when transitioning between subtasks, assigning lower rewards in later phases, which causes the agent to become stuck. In contrast, REDS, guided by the EPIC loss, produces consistent and context-aware reward signals, enabling the agent to progress effectively through subsequent subtasks.
>
> ---
>
> **Reference**\
> [1] VIP: Towards Universal Visual Reward and Representation via Value-Implicit Pre-Training, ICLR 2023.\
> [2] XIRL: Cross-embodiment Inverse Reinforcement Learning, CoRL 2021.\
> [3] Unsupervised Perceptual Rewards for Imitation Learning, RSS 2017.\
> [4] MimicGen: A Data Generation System for Scalable Robot Learning using Human Demonstrations, CoRL 2023.\
> [5] Learning multi-stage tasks with one demonstration via self-replay, CoRL 2021.\
> [6] Visual Language Maps for Robot Navigation, ICRA 2023.\
> [7] Octopus: Embodied Vision-Language Programmer from Environmental Feedback, ECCV 2024.\
> [8] Generative Agents: Interactive Simulacra of Human Behavior, UIST 2023.\
> [9] Mobile-Agent-v2: Mobile Device Operation Assistant with Effective Navigation via Multi-Agent Collaboration, NeurIPS 2024.\
> [10] Language-Grounded Dynamic Scene Graphs for Interactive Object Search with Mobile Manipulation, RA-L 2024.\
> [11] BUMBLE: Unifying Reasoning and Acting with Vision-Language Models for Building-wide Mobile Manipulation, ArXiv 2024.\
> [12] FurnitureBench: Reproducible Real-World Benchmark for Long-Horizon Complex Manipulation, RSS 2023.\
> [13] Diffusion Reward: Learning Rewards via Conditional Video Diffusion, ECCV 2024.

---

> > ### Comment · Reviewer_VAFb · 2024-11-22
> >
> > Thanks for the detailed responses and additional results. Most of my concerns are well addressed. So, I raise my rating to 'accept'. Good luck with your real-world experiments, and I wish to see REDS work on more complex real-world robot tasks.

---

> > > ### Author Response · Authors · 2024-11-22
> > >
> > > Dear Reviewer VAFb,
> > >
> > > Thank you for your response. We are happy to hear that we have addressed most of your concerns.
> > > Following your suggestion, we will include experimental results on more complex real-world robot tasks in the final manuscript.
> > > If you have any further questions or suggestions, please do not hesitate to let us know.
> > >
> > > Thank you very much!\
> > > Authors

---

### Official Review · Reviewer_Vi9Z · 2024-11-01

**Soundness:** 2
**Presentation:** 2
**Contribution:** 2
**Rating:** 6
**Confidence:** 4

**Summary:**

This paper introduces an inverse reinforcement learning approach that enables the extraction of a dense reward signal from video demonstrations with minimal human intervention. The reward model is designed to consider both the observation and subtask information. This connection between the reward and subtask information intuitively breaks down the intricate long-term task into manageable subtasks. Empirical findings further validate the benefits of integrating subtask information into the reward learning process.

**Strengths:**

i) In this paper, subtask information is integrated into the reward learning process. During training, the subtask is included in the reward function input as a text embedding that provides instructions on completing a specific subtask. In the inference phase, the text embedding is substituted with a video embedding as an additional input.

ii) The approach employs the EPIC loss function to reduce the disparity between the predicted reward sequence and the ground truth reward. Experimental results demonstrate superiority of this approach.

**Weaknesses:**

i) In the training phase, this paper decomposes the overall task into multiple subtasks based on domain knowledge. However, the reliance on predefined instructions from the environment for task decomposition raises concerns about practical applicability. Some environments may lack such predefined knowledge, necessitating human annotations or the need for a learned task decomposition model when extending this approach to new environments. This raises doubts about the novelty and scalability of this methodology.

ii) This paper introduces the function $\psi$, which maps observations to subtasks. However, providing more details about this function is essential. Specifically, elucidating the form of the subtask output by $\psi$ is crucial since it serves as the ground truth for reward learning, a key component for the success of this method. Therefore, the authors should elaborate on the detail of the subtasks produced by $\psi$.

iii) During the reward learning, this method utilizes the EPIC loss function to quantify the disparity between the estimated reward and the proposed ground truth. While EPIC was initially introduced in a prior work, the rationale behind its selection and comparisons with previous loss functions for reward learning should be further discussed by the authors. This explanation should include both intuitive reasoning and empirical evidence.

iv) During the inference phase, the substitution of text embedding with video embedding for subtask information is addressed. Although an alignment mechanism is employed to bring video and text embeddings closer in the latent space, the loss function appears to overlook the maximization of distances between irrelevant embeddings. This oversight could be crucial when dealing with subtasks that share similar text descriptions or video content. It is advisable for the authors to investigate the effectiveness of learned embeddings within tasks involving similar subtasks, although this aspect is not a primary concern.

**Questions:**

What if we utilize the text embedding directly as input during the inference phase? Does this approach exhibit a substantial performance advantage over the method that employs video embedding?

---

> ### Author Response · Authors · 2024-11-21
>
> Dear Reviewer Vi9Z,
>
> We sincerely appreciate your valuable comments, which were extremely helpful in improving our draft. Below, we address each comment in detail.
>
> ---
>
> **[W1] Reliance on domain-based task decomposition**
>
> Thank you for pointing this out. In defining subtasks, we designed a rule that avoids manual, task-specific adjustments and broadly applies to robotic manipulation tasks. To this end, we assumed that tasks can be decomposed into a sequence of object-centric subtasks, where each subtask involves manipulation relative to a single object, following prior work [1, 2]. For instance, in Door Open (Figure 2a of the draft), the subtasks are (i) reaching the door handle (motion relative to the handle) and (ii) pulling the door to the goal position (motion relative to a green sphere). This approach is intuitive because humans naturally perceive tasks as sequences of discrete object interactions, each with a clear and measurable goal. For new tasks, subtasks can be defined by identifying the manipulated object and its goal state, following the same objective criteria. This ensures that the method is consistent, repeatable, and easily generalizable across different tasks.
>
> To further address concerns about scalability and reliance on predefined instructions, we plan to integrate ideas from concurrent works, generating high-level plans with instructions for solving each subtask and specifying current progress using the reasoning capability of Multimodal Large Language models (MLLM) [3] or decomposing long-horizon robotics demonstrations into subtasks using pre-trained vision language models [4, 5], in the future work. We have added an in-depth discussion of these extensions in the revised manuscript.
>
> **[W2] Clarification of subtask identification function $\psi$**
>
> Thank you for your detailed question. $\psi$ maps observations at each timestep to the index of the ongoing subtask, incrementing its output as subtasks are completed. This results in the form of a step function, where each step corresponds to the completion of a subtask. Please refer to the graph in the center of Figure 1 in the revised draft for visual examples.
>
> **[W3] Rationale and comparisons for EPIC loss selection in reward learning**
>
> As demonstrated in prior work [6], a low EPIC distance between a learned reward $\hat{R}$ and the true reward $R$ predicts low regret, which implies that policies $\pi_{\hat{R}}$​ and $\pi_R$, optimized for $\hat{R}$ and $R$, respectively, achieve similar returns under the ground-truth reward $R$. Leveraging this insight, we adopt EPIC distance as an optimization objective to train a dense reward function that induces the same set of optimal policies as $\psi$, ensuring robust and consistent performance across tasks.
>
> For empirical comparison, we provide qualitative analyses of different reward functions in Figure 10. Discriminator-based methods are observed to suffer from mode collapse, leading to suboptimal reward signals. Similarly, video prediction-based approaches tend to assign lower rewards to later subtasks, causing agents to stagnate in earlier phases of the task. In contrast, trained with EPIC loss, REDS produces consistent and task-progressive reward signals, allowing agents to transition effectively through subtasks. These results demonstrate our approach's robustness in addressing prior methods' limitations.
>
> **[W4] Dealing with subtasks sharing similar text descriptions or video contents**
>
> Thank you for your constructive feedback. Section 5.4 demonstrates REDS's generalization capabilities in unseen environments requiring similar subtasks with unseen objects (Window Close), where the text instructions differ only by the target object. Figure 5 shows that REDS generates effective reward signals and achieves comparable or superior RL performance, even under these conditions. While the alignment mechanism currently focuses on minimizing distances between relevant embeddings, we acknowledge that incorporating a loss term to maximize distances between irrelevant embeddings could further improve robustness in tasks with similar subtasks. We will explore this enhancement in future work and have added a discussion of this limitation in the revised manuscript.

---

> > ### Author Response · Authors · 2024-11-21
> >
> > **[Q1] Using text embedding directly during the inference phase**
> >
> > We would like to clarify that we use both video and text embedding of the inferred subtask as input for generating rewards in online interaction. With this design choice, REDS can produce context-aware reward signals according to the inferred subtasks in online interactions. We included this clarification in the revised manuscript.
> >
> > ---
> >
> > **Reference**\
> > [1] MimicGen: A Data Generation System for Scalable Robot Learning using Human Demonstrations, CoRL 2023.\
> > [2] Learning multi-stage tasks with one demonstration via self-replay, CoRL 2022.\
> > [3] Robotic Control via Embodied Chain-of-Thought Reasoning, CoRL 2024.\
> > [4] KISA: A Unified Keyframe Identifier and Skill Annotator for Long-Horizon Robotics Demonstrations, ICML 2024.\
> > [5] Universal Visual Decomposer: Long-Horizon Manipulation Made Easy, ICRA 2024.\
> > [6] Quantifying Differences in Reward Functions, ICLR 2022.

---

> > ### Comment · Reviewer_Vi9Z · 2024-11-29
> >
> > Thanks for authors' responses.
> > W1. I have checked the previous work [1] that involves the task decomposition, as suggested in your paper. This work used a stage indicator to determine whether the agent has entered a stage (subtask). And this stage indicator is defined in the environment and is assumed to be accessible to the agent. Thus, I am wondering if you also use this type of stage indicator for the task decomposition. If so, I have to say this task decomposition method still needs expertise knowledge and human intervention.
> >
> > W2-W4. I acknowledge that the responses have addressed my concerns regarding W2, W3 and W4.
> >
> > [1] DrS: Learning Reusable Dense Rewards for Multi-Stage Tasks. ICLR2024

---

> > > ### Author Response · Authors · 2024-11-30
> > > **Response to Reviewer Vi9Z**
> > >
> > > Dear Reviewer Vi9Z,
> > >
> > > Thank you for your thoughtful feedback. We are pleased that our responses addressed your concerns regarding W2, W3, and W4. We also appreciate the opportunity to clarify the distinction between our method, REDS, and DrS [1].
> > >
> > > We would like to emphasize that REDS does not rely on stage indicators or require human intervention during online interactions. Instead, it autonomously infers ongoing subtasks from visual observations and generates reward signals accordingly. Importantly, REDS only requires subtask segmentation for expert demonstrations, guided by intuitive and generalizable object-centric subtask rules before the training loop. Once training begins, no additional human input is necessary, enabling fully autonomous training, as demonstrated in Section 5.2.
> > >
> > > In contrast, DrS depends on subtask indicators defined within the environment, requiring manual annotations from human experts during every online interaction. This reliance poses a significant bottleneck for scaling real-world applications. Moreover, DrS necessitates subtask annotations for suboptimal demonstrations, further increasing the burden of manual segmentation, which can be error-prone and labor-intensive, hindering its scalability.
> > >
> > > We believe this clarification highlights the strengths of our approach and addresses concerns. REDS provides a significant advantage over prior work by mitigating reliance on human intervention and enabling scalable, fully autonomous training. We respectfully request that you reconsider your evaluation in light of these distinctions.
> > >
> > > Thank you again for your thoughtful review and for helping improve our work. If you have any further questions or suggestions, please do not hesitate to let us know.
> > >
> > > Many Thanks,\
> > > Authors
> > >
> > > **References**\
> > > [1] DrS: Learning Reusable Dense Rewards for Multi-Stage Tasks, ICLR2024.

---

> > > > ### Comment · Reviewer_Vi9Z · 2024-11-30
> > > >
> > > > The authors have introduced the details for both REDS and DrS. But I am still confused about the implementation of the "task decomposition" in your work. You claimed that you have designed a task-agnostic rule for the task decomposition. Thus, I also checked the paper you cited in the rebuttal. In [1], a stage-recognition network is trained to  predicts the stage of the task the robot is currently in. I am wondering if you also used this network to complete the task decomposition.
> > > >
> > > >
> > > > [1] Learning multi-stage tasks with one demonstration via self-replay.

---

> ### Author Response · Authors · 2024-11-24
> **Gentle Reminder**
>
> Dear Reviewer Vi9Z,
>
> Thank you again for your time and efforts in reviewing our paper.
>
> As the discussion period draws close, we kindly remind you that two days remain for further comments or questions. We would appreciate the opportunity to address any additional concerns you may have before the discussion phase ends.
>
> Thank you very much!
>
> Many thanks,\
> Authors

---

> ### Author Response · Authors · 2024-11-30
> **Response to Reviewer Vi9Z**
>
> [Training Phase]\
> We begin by decomposing the task into a sequence of object-centric subtasks, where each subtask involves manipulation relative to a single object (as detailed in Section 4.1 of the revised draft). Expert demonstrations are segmented to label ongoing subtasks from the sequence of object-centric subtasks at each timestep. These segmentations are used to train REDS with the objectives described in Section 4, including a contrastive learning objective that aligns video embeddings with their corresponding subtask embeddings. Notably, no additional segmentation is required once training begins.
>
> [Inference Phase]\
> Although our approach aligns with the overarching goal of [1], REDS implements a distinct mechanism for task decomposition. Instead of training a stage-recognition network, REDS infers subtasks during online interaction by comparing video and text embeddings. Specifically, a video encoder processes the observation history, while task instructions for all subtasks are pre-encoded using a text encoder. The ongoing subtask $i$ is identified by selecting the subtask embedding $\mathbf{e}_{i}$ with the highest cosine similarity to the video embedding. This process is made robust by training video representations to align closely with subtask embeddings via the contrastive learning objective (Equation 5 in the revised draft). This alignment significantly enhances subtask inference and improves RL performance, as demonstrated in Figure 9a.
>
> Please let us know if further clarification would be helpful.
>
> **References**\
> [1] Learning multi-stage tasks with one demonstration via self-replay, CoRL 2021.

---

> > ### Comment · Reviewer_Vi9Z · 2024-11-30
> >
> > Thanks for your responses. I checked the section 4.1 in the revised draft. In the last sentence of section 4.1, you mentioned that you use the predefined codes in Meta-world and human annotators to obtain subtask segmentation as the label for the training. Please highlight that you use the network to predict the subtask that is trained using the labelled subtask segmentation.

---

> > > ### Author Response · Authors · 2024-11-30
> > > **Response to Reviewer Vi9Z**
> > >
> > > Thank you for your thoughtful feedback and valuable suggestions. As the period for uploading a revised PDF has passed, we will definitely incorporate your suggestion in the camera-ready version by explicitly highlighting how the network predicts subtasks during online interactions. Specifically, we will clarify that annotating subtask segmentation labels is only for training, and our training scheme enables the network to infer subtasks automatically during online interactions.
> > >
> > > We hope this clarification demonstrates our attentiveness to your concerns and the strength of our approach. If our response adequately addresses your concerns, we would sincerely appreciate your consideration in raising the score. Please let us know if you have additional suggestions or areas where further clarification would be helpful. **Your perspective is greatly valued, and we look forward to your response.**

---

> > > > ### Comment · Reviewer_Vi9Z · 2024-11-30
> > > >
> > > > Thanks for your effort in the rebuttal phase. I am inclined to raise the score from 5 to 6.

---

> ### Author Response · Authors · 2024-11-30
> **Response to Reviewer Vi9Z**
>
> Thank you for your response. We are happy to hear that we have addressed most of your concerns. Following your suggestion, we will further clarify the advantages of our method in the final manuscript. Again, Thank you for the valuable suggestion and your positive assessment of our work.
>
> Thank you very much!\
> Authors

---

### Official Review · Reviewer_j9z4 · 2024-11-04

**Soundness:** 2
**Presentation:** 3
**Contribution:** 2
**Rating:** 5
**Confidence:** 3

**Summary:**

This paper addresses the challenge in reinforcement learning (RL) of relying heavily on human-designed reward functions, especially for long-horizon tasks. It introduces REDS (Reward Learning from Demonstration with Segmentations), which infers subtask information from video segments and generates corresponding reward signals for each subtask, utilizing minimal supervision. The framework employs contrastive learning objectives to align video representations with subtasks and uses the EquivalentPolicy Invariant Comparison (EPIC) distance to minimize the difference between the learned reward function and ground-truth rewards.

**Strengths:**

1. This paper is written clearly and highlights an important challenge for long-horizon reinforcement learning.
2. This paper provides a reward model training method that utilizes both expert demonstration videos and suboptimal videos.

**Weaknesses:**

- The method seems to rely heavily on carefully predefined subtasks or key completion points (like in table 6). This may limit the generalizability of the method.

**Questions:**

1. Note that in the experimental setting of this article, expert demonstration videos of the current task are already available. Should imitation-learning-from-observation-based methods also be used as a baseline, for example, to directly derive rewards based on the similarity between the agent trajectory and the expert demonstration video?
2. When learning a new task, how can we determine the appropriate way to divide it into subtasks? For instance, should it be broken down into 2, 3, or 4 subtasks?

---

> ### Author Response · Authors · 2024-11-21
>
> Dear Reviewer j9z4,
>
> We sincerely appreciate your valuable comments, which were extremely helpful in improving our draft. Below, we address each comment in detail.
>
> ---
>
> **[W1, Q2] Explanation of subtask definition**
>
> Thank you for pointing this out. In defining subtasks, we designed a rule that avoids manual, task-specific adjustments and broadly applies to robotic manipulation tasks. To this end, we assumed that tasks can be decomposed into a sequence of object-centric subtasks, where each subtask involves manipulation relative to a single object, following prior work [1, 2]. For instance, in Door Open (Figure 2a of the draft), the subtasks are (i) reaching the door handle (motion relative to the handle) and (ii) pulling the door to the goal position (motion relative to a green sphere). This approach is intuitive because humans naturally perceive tasks as sequences of discrete object interactions, each with a clear and measurable goal. Furthermore, this assumption can be generally applied to different manipulation skills (e.g., pick-and-place, inserting, assembling) with diverse objects.
>
> For new tasks, subtasks can be defined by identifying the manipulated object and its goal state, following the same objective criteria. This ensures that the method is consistent, repeatable, and easily generalizable across different tasks. In the revised manuscript, we have elaborated on the rules for subtask definition.
>
> **[Q1] Using imitation-from-observation methods as a baseline**
>
> We would like to clarify that several of our baselines, including ORIL, DrS, and R2R, are variants of discriminator-based reward learning methods. These approaches train a discriminator to differentiate between expert demonstrations and agent trajectories, using the discriminator's output as a reward signal based on the trajectory's alignment with expert behavior. REDS consistently and significantly outperforms these baselines in our experiments across all tasks.
>
> ---
> **Reference**\
> [1] MimicGen: A Data Generation System for Scalable Robot Learning using Human Demonstrations, CoRL 2023.\
> [2] Learning multi-stage tasks with one demonstration via self-replay, CoRL 2022.

---

> ### Author Response · Authors · 2024-11-24
> **Gentle Reminder**
>
> Dear Reviewer j9z4,
>
> Thank you again for your time and efforts in reviewing our paper.
>
> As the discussion period draws close, we kindly remind you that two days remain for further comments or questions. We would appreciate the opportunity to address any additional concerns you may have before the discussion phase ends.
>
> Thank you very much!
>
> Many thanks,\
> Authors

---

> ### Comment · Reviewer_j9z4 · 2024-11-26
>
> Thanks for the author's reply. After reading the reply, my concern still holds that the subtask definition heavily relies on human annotations. As a result, I decide to keep the score unchanged.

---

> > ### Author Response · Authors · 2024-11-27
> > **Response to Reviewer j9z4**
> >
> > Dear Reviewer j9z4,
> >
> > Thank you for your thoughtful feedback. We appreciate the opportunity to address your concern regarding the reliance on human annotations for subtask definitions.
> >
> > REDS is explicitly designed to minimize reliance on human annotations by significantly reducing the manual effort involved in reward design. Unlike conventional approaches, which often require extensive human workloads and numerous trial-and-errors, REDS relies on a one-time specification of generalizable object-centric subtask rules, thereby alleviating the need for repetitive human input.
> >
> > Furthermore, this one-time specification can be automated by other AI models. Recent advancements, such as the use of Multimodal Large Language Models (MLLM) for automating subtask definitions and checking task progress [1], as well as keyframe extraction from long-horizon trajectories using reconstruction errors [2], can provide alternative sources for subtask definitions. Since REDS is agnostic to the source of subtask definitions—whether derived from human annotations, MLLMs, or automated methods—it seamlessly integrates with these techniques, further minimizing manual effort.
> >
> > To summarize:
> >
> > - Subtask definitions in REDS are not limited to human annotations. Automated sources, such as MLLMs or keyframe extraction, are equally applicable.
> > - REDS is designed to integrate with diverse sources of subtask definitions, making it a versatile and scalable framework for reward design.
> >
> > We believe this explanation clarifies how our work effectively addresses the concern about reliance on human annotations while aligning with recent advancements in automation. We kindly ask you to reconsider your evaluation in light of these points.
> >
> > Many Thanks,\
> > Authors
> >
> > **Reference**\
> > [1] Robotic Control via Embodied Chain-of-Thought Reasoning, CoRL 2024.\
> > [2] Waypoint-Based Imitation Learning for Robotic Manipulation, CoRL 2024.

---

> > > ### Author Response · Authors · 2024-12-01
> > > **Gentle Reminder**
> > >
> > > Dear Reviewer j9z4,
> > >
> > > Thank you again for your time and efforts in reviewing our paper.
> > >
> > > As the discussion period draws close, we kindly remind you that two days remain for further comments or questions. We hope to confirm that we’ve adequately addressed your concerns and are open to discussing any remaining points or questions you may have.  If our response adequately addresses your concerns, we sincerely appreciate your consideration in raising the score. **Your input is invaluable to us, and we would greatly value the chance to discuss them with you.**
> > >
> > > Thank you very much!
> > >
> > > Many thanks,\
> > > Authors

---

### Official Review · Reviewer_mmyJ · 2024-11-05

**Soundness:** 3
**Presentation:** 3
**Contribution:** 3
**Rating:** 6
**Confidence:** 3

**Summary:**

Demonstration with Segmentations (REDS), for long-horizon robotic tasks that are usually composed of numerous subtasks. The key intuition is to use subtask annotations or descriptions for expert demonstrations to aid in learning a dense reward function. The reasoning behind this approach is that the subtasks occur in a particular order in expert demonstrations, and this order can be used as ground truth to learn the reward function.

The reward model or the reward predictor is defined as a function of the sequence of prior observations and the current subtask. The model is trained using the Equivalent-Policy Invariant Comparison (EPIC) metric. The goal is to match the learned reward function to the ground-truth reward function obtained from the ordering of subtasks in expert demonstrations. To identify the current subtask, the paper uses the cosine similarity between the CLIP image embeddings of the observations and the CLIP text embeddings of the textual descriptions of the subtasks required to achieve the goal. Therefore, before the execution of an episode, the subtasks have to be defined in natural language.

Additionally, the authors propose to continually learn the reward function from suboptimal demonstrations obtained from an RL agent. Here, the RL agent is trained using the reward function obtained from expert demonstrations. This process is repeated multiple times.

 The proposed approach outperforms other reward learning baselines for manipulation tasks from the Meta-world and the FurnitureBench environments.

**Strengths:**

Overall, the paper is well written, and the problem is very well explained. The authors have clearly delineated their contributions from previous works. The experiments and ablations are detailed and informative. With respect to the proposed approach, the strengths are:
The approach requires minimal human supervision in terms of defining the subtasks accurately.
The proposed approach generalizes well for manipulation tasks with unseen objects. Additionally, since the reward model does not depend on actions, the authors show generalization capabilities to different robots. For example, the reward model trained only with the Panda arm can be used to train RL agents for the Sawyer arm.
Dependence on foundation models like CLIP for subtask identification reduces the adversarial effects of visual artifacts like varying lighting conditions, locations of objects, etc.

**Weaknesses:**

The expert demonstrations would always contain the subtasks in a particular order. This might lead to poor reward signals when the subtask estimation turns out to be incorrect. Such instances could occur when the RL agent is exploring.
The effect of the hyperparameter epsilon, which is used to enforce progressive reward signals within each subtask, is not clearly explained. The authors show an ablation for the cases with and without regularization loss. But, the effect of epsilon is not clearly described.
From Fig. 3, it appears that the baselines have not converged for the Lever Pull, Sweep Into, and Push cases.
The proposed approach is not compared against CLIP-based zero-shot reward models like [1].
LIV [2] also learns dense rewards from just observations and text descriptions. The proposed approach has to be either differentiated from or compared against this paper.

[1] Vision-Language Models are Zero-Shot Reward Models for Reinforcement Learning.
[2] LIV: Language-Image Representations and Rewards for Robotic Control.

**Questions:**

Questions:
Is there a way to quantify the subtask identification performance? This could be useful as the identified subtask affects the reward prediction.
An ablation study depicting the effect of epsilon, which is used to enforce progressive reward signals, will be useful. CLIP-based subtask identification and ground-truth rewards from subtask ordering are not sufficient to enforce the required expert behavior. From my understanding, this is implicitly enforced using progressive reward signals. Therefore, an ablation study for epsilon will be crucial.
The authors need to explain how their approach is different from [1] and [2].

[1] Vision-Language Models are Zero-Shot Reward Models for Reinforcement Learning.
[2] LIV: Language-Image Representations and Rewards for Robotic Control.

---

> ### Author Response · Authors · 2024-11-21
>
> Dear Reviewer mmyJ,
>
> We sincerely appreciate your valuable comments, which were extremely helpful in improving our draft. Below, we address each comment in detail.
>
> ---
>
> **[W1, Q1] Relation between subtask identification and RL performance**
>
> We agree that the reward model trained only with expert demonstrations might produce unreliable reward signals with incorrect subtask identification in the exploration phase. To solve this issue, as mentioned in Section 4.4, We fine-tuned our reward model using additional suboptimal demonstrations obtained from RL agents initially trained with expert demonstrations for preventing reward misspecification.
>
> To quantify the effect of fine-tuning, we measure the accuracy of the reward model's subtask identification using unseen datasets before and after fine-tuning with additional suboptimal demonstrations in the revised draft. Table 7 of the revised draft demonstrates that fine-tuning notably enhances precision on suboptimal demonstrations, thereby improving RL performance.
>
> **[W2, Q2] Effect of hyperparameter $\epsilon$**
>
> For choosing the hyper-parameter $\epsilon$, which enforces progressive reward signals, we conducted a grid search in the range [0.0, 1.0] and selected 0.5 for all experiments. We included new RL performance results across different $\epsilon$ values in Figure 12 of the revised draft.  We observe that smaller values failed to provide sufficient progressive signals, while larger values reduced the accuracy of subtask inference, degrading the reward function's overall effectiveness.
>
>
> **[W3] Convergence of baselines on some tasks**
>
> Thank you for your pointer! To address this concern, we extended the training for RL agents up to 5 million environmental interactions for the Push and Coffee Pull tasks. We have updated the results (see Figure 1) in the revised draft. Despite the increased training interactions, we observed that the overall performance trends remained consistent with our initial findings.
>
> **[W4, Q4] Comparison with prior work [1,2]**
>
> Both VLM-RM [1] and LIV [2] share similarities with REDS in terms of producing dense rewards from visual observations and text instructions. However, REDS distinguishes itself by its ability to infer ongoing subtasks and generate dynamic, context-aware reward signals tailored to specific text instructions during online interactions. In contrast, VLM-RM and LIV require static text instructions describing the overall task or depend on external mechanisms for subtask identification. We have included a comparison in Figure 12 of the revised draft, where RL agents trained using REDS significantly outperform those using VLM-RM and LIV, demonstrating the effectiveness of our approach.
>
> ---
>
> **References**\
> [1] Vision-Language Models are Zero-Shot Reward Models for Reinforcement Learning, ICLR 2024.\
> [2] LIV: Language-Image Representations and Rewards for Robotic Control, ICML 2023.

---

> ### Author Response · Authors · 2024-11-24
> **Gentle Reminder**
>
> Dear Reviewer mmyJ,
>
> Thank you again for your time and efforts in reviewing our paper.
>
> As the discussion period draws close, we kindly remind you that two days remain for further comments or questions. We would appreciate the opportunity to address any additional concerns you may have before the discussion phase ends.
>
> Thank you very much!
>
> Many thanks,\
> Authors

---

### Author Response · Authors · 2024-11-21
**Summary of Revisions**

We deeply appreciate your time and effort in reviewing our manuscript. As the reviewers highlighted, we propose a simple but effective reward learning framework (all reviewers) for long-horizon complex robotic manipulation tasks, which is an important challenge (j9z4). Our approach utilizes action-free videos and requires minimal human efforts (mmyJ) without hand-engineering (mmyJ). We demonstrate REDS's strong empirical performance (all reviewers) in complex tasks and provide comprehensive ablation studies (j9z4, VAFb) to support the claim with a clear presentation (mmyJ, j9z4, VAFb).

We appreciate the reviewers’ insightful comments on our manuscript. In response to the questions and concerns you raised, we have carefully revised and enhanced the manuscript with the following additional experiments and discussions:

- Clarification on motivation (Section 1)
- Improving the method description (Section 4.1, Section 4.4, Appendix A)
- Additional discussion on limitations and future direction (Section LIMITATION AND FUTURE DIRECTIONS)
- Additional experimental results (Appendix G)
  - Comparison with additional baseline (Diffusion Reward, CLIP, LIV)
  - Investigating the effect of $\epsilon$ scaling progressive reward signals
  - Investigating the effect of $T_{U}$ for subtask identification
  - Quantifying precision on subtask identification

These updates are temporarily highlighted in "cyan" for your convenience to check.\
We strongly believe that REDS can be a useful addition to the ICLR community, particularly because reviewers’ constructive comments enhanced the manuscript.

Thank you very much,\
Authors

---

### Meta-Review · Area_Chair_x8nj · 2024-12-19

**Metareview:**

The paper addresses a challenge in reinforcement learning, specifically the heavy reliance on human-designed reward functions for long-horizon tasks. The proposed REDS framework offers a new solution by leveraging segmented video demonstrations with minimal supervision to learn reward functions. The key innovation of aligning video segments with subtasks using contrastive learning and the EPIC distance has been well-explained and is supported by experimental results. The experiments are extensive, covering both simulated (Meta-World) and real-world (FurnitureBench) tasks. The results demonstrate the superiority of REDS in long-horizon tasks with multiple subtasks, which is a significant contribution. The ablation studies and comparisons with various baselines provide strong evidence of the method's effectiveness.

The overall assessment of the paper is positive, with multiple reviewers acknowledging its strengths. The authors have effectively addressed the concerns raised by the reviewers in their rebuttals and subsequent discussions. While some minor concerns remain, such as the potential for further improvement in handling subtle visual changes and the need for more complex real-world experiments, the authors have demonstrated a clear understanding of these issues and have plans to address them in future work. Considering the overall strengths of the paper, the effective responses to reviewer comments, and the potential, the paper is worthy of acceptance.

**Additional Comments On Reviewer Discussion:**

The primary concerns regarding this work center around the dependence on human annotations for both the definition of subtasks and the collection of subtask segmentations. During the rebuttal, the authors emphasized that the definitions within REDS are not exclusively bound to human annotations; automated sources can be equally valid. Moreover, the methodology employed is indifferent to the origin of the subtask definitions. The reviews also presented additional inquiries concerning different aspects, all of which have been satisfactorily addressed during the rebuttal.

---

### Decision · Program_Chairs · 2025-01-22

Accept (Poster)